# ReVar: Strengthening Policy Evaluation via Reduced Variance Sampling

**Subhojyoti Mukherjee**[1]        **Josiah P. Hanna**[2]        **Robert Nowak**[1]

[1]Department of Electrical and Computer Engineering, University of Wisconsin-Madison, USA
[2]Computer Sciences Department, University of Wisconsin-Madison, USA

## Abstract

This paper studies the problem of data collection for policy evaluation in Markov decision processes (MDPs). In policy evaluation, we are given a *target* policy and asked to estimate the expected cumulative reward it will obtain in an environment formalized as an MDP. We develop theory for optimal data collection within the class of tree-structured MDPs by first deriving an oracle data collection strategy that uses knowledge of the variance of the reward distributions. We then introduce the **Re**duced **Var**iance Sampling (ReVar) algorithm that approximates the oracle strategy when the reward variances are unknown a priori and bound its sub-optimality compared to the oracle strategy. Finally, we empirically validate that ReVar leads to policy evaluation with mean squared error comparable to the oracle strategy and significantly lower than simply running the target policy.

## 1 INTRODUCTION

In reinforcement learning (RL) applications, there is often a need for policy evaluation to determine (or estimate) the expected return (future cumulative reward) of a given policy. Policy evaluation is also required in other sequential decision-making settings outside of RL. For example, testing an autonomous vehicle stack or ad-serving system can be seen as policy evaluation applications. Accurate and data efficient policy evaluation is critical for safe and trust-worthy deployment of autonomous systems.

This paper studies data collection for low mean squared error (MSE) policy evaluation in sequential decision-making tasks formalized as Markov decision processes (MDPs). The objective of policy evaluation is to estimate the expected return that will be obtained by running a *target policy* which is a given probabilistic mapping from states to actions.

To evaluate the target policy, we require data from the environment in which it will be deployed. Collecting data requires running a (possibly non-stationary) *behavior* policy to generate state-action-reward trajectories. Our goal is to find a behavior policy that leads to a minimum MSE evaluation of the target policy.

The most natural choice is *on-policy sampling* in which we use the target policy as the behavior policy. However, we show that in some cases this choice is far from optimal (e.g., Figure 2 in our empirical analysis) as it fails to actively take actions from which the expected return is uncertain. Instead, an optimal behavior policy should take actions in any given state to reduce uncertainty in the current estimate of the expected return from that state.

Our paper makes the following main contributions. We first derive an optimal "oracle" behavior policy for finite tree-structured MDPs *assuming oracle access to the MDP transition probabilities and variances of the reward distributions*. Sampling trajectories according to the oracle behavior policy minimizes the MSE of the estimator of the target policy's expected. As a special case (depth 1 tree MDPs), we recover the optimal behavior policy for multi-armed bandits Carpentier et al. [2015].

We then introduce a practical algorithm, **Re**duced **Var**iance Sampling (ReVar), that adaptively learns the optimal behavior policy by observing rewards and adjusting the policy to select actions that reduce the MSE of the estimator. The main idea of ReVar is to plug-in upper-confidence bounds on the reward distribution variances to approximate the oracle behavior policy. We define a notion of policy evaluation regret compared to the oracle behavior policy, and bound the regret of ReVar. The regret converges rapidly to $0$ as the number of sampled episodes grows, theoretically guaranteeing that ReVar quickly matches the performance of the oracle policy. Finally, we implement ReVar and show it leads to low MSE policy evaluation in both a tree-structured and a general finite-horizon MDP. Taken together, our contributions provide a theoretical foundation towards optimal

*Accepted for the 38th Conference on Uncertainty in Artificial Intelligence* (UAI 2022).

data collection for policy evaluation in MDPs.

The remainder of the paper is organized as follows. In Section 3 we reformulate our problem in the bandit setting and discuss related bandit works. In Section 4 we extend the bandit formulation to the tree MDP. Finally we introduce the more general Directed Acyclic Graph (DAG) MDP in Section 5 and discuss some limitations of our sampling behavior. We show numerical experiments in Section 6 and conclude in Section 7.

## 2 BACKGROUND

In this section, we introduce notation, define the policy evaluation problem, and discuss the prior literature.

### 2.1 NOTATION

A finite-horizon Markov Decision Process, $\mathbf{M}$, is the tuple $(\mathcal{S}, \mathcal{A}, P, R, \gamma, d_0, L)$, where $\mathcal{S}$ is a finite set of states, $\mathcal{A}$ is a finite set of actions, $P : \mathcal{S} \times \mathcal{A} \times \mathcal{S} \to [0,1]$ is a state transition function, $R$ is the reward distribution (formalized below), $\gamma \in [0,1)$ is the discount factor, $d_0$ is the starting state distribution, and $L$ is the maximum episode length. A (stationary) policy, $\pi : \mathcal{S} \times \mathcal{A} \to [0,1]$, is a probability distribution over actions conditioned on a given state. We assume data can only be collected through episodic interaction: an agent begins in state $S_0 \sim d_0$ and then at each step $t$ takes an action $A_t \sim \pi(\cdot|S_t)$ and proceeds to state $S_{t+1} \sim P(\cdot|S_t, A_t)$. Interaction terminates in at most $L$ steps. Each time the agent takes action $a_t$ in state $s_t$ it observes a reward $R_t \sim R(s_t, a_t)$. We assume $R(s,a) = \mathcal{P}(\mu(s,a), \sigma^2(s,a))$, where $\mathcal{P}$ denotes a parametric distribution with mean $\mu(s,a)$ and variance $\sigma^2(s,a)$. The entire interaction produces a trajectory $H := \{(S_t, A_t, R_t)\}_{t=1}^L$. We assume $d_0$ is known but $P$ and the reward distributions are unknown. We define the value of a policy as: $v(\pi) := \mathbb{E}_\pi[\sum_{t=1}^L \gamma^{t-1} R_t]$, where $\mathbb{E}_\pi$ is the expectation w.r.t. trajectories sampled by following $\pi$.

We will make use of the fact that the value of a policy can be written as: $v(\pi) = \mathbb{E}[v_0^\pi(S_0)|S_0 \sim d_0]$ where,

$$v_t^\pi(s) := \sum_a \pi(a|s)\mu(s,a) + \gamma \sum_{s'} P(s'|s,a)v_{t+1}^\pi(s')$$

for $t \leq L$ and $v_t^\pi(s) = 0$ for $t > L$.

### 2.2 POLICY EVALUATION

We now formally define our objective. We are given a target policy, $\pi$, for which we want to estimate $v(\pi)$. To estimate $v(\pi)$ we will generate a set of $K$ trajectories where each trajectory is generated by following some policy. Let $H^k := \{s_t^k, a_t^k, R_t^k(s_t^k, a_t^k)\}_{t=1}^L$ be the trajectory collected in episode $k$ and let $b^k$ be the policy ran to produce $H^k$. The entire set of collected data is given as $\mathcal{D} := \{H^k, b^k\}_{k=1}^K$.

Once $\mathcal{D}$ is collected, we estimate $v(\pi)$ with a certainty-equivalence estimate Sutton [1988]. Suppose $\mathcal{D}$ consists of $n = KL$ state-action transitions. We define the random variable representing the estimated future reward from state $s$ at time-step $t$ as:

$$Y_n(s,t) := \sum_a \pi(a|s)\widehat{\mu}(s,a) + \gamma \sum_{s'} \widehat{P}(s'|s,a)Y_n(s',t+1),$$

where $Y_n(s, t+1) := 0$ if $t \geq L$, $\widehat{\mu}(s,a)$ is an estimate of $\mu(s,a)$ and $\widehat{P}(s'|s,a)$ is an estimate of $P(s'|s,a)$, both computed from $\mathcal{D}$. Finally, the estimate of $v(\pi)$ is computed as $Y_n := \sum_s d_0(s)Y_n(s,0)$. In the policy evaluation literature, the certainty-equivalence estimator is also known as the direct method Jiang and Li [2016] and, in tabular settings, can be shown to be equivalent to batch temporal-difference estimators Sutton [1988], Pavse et al. [2020]. Thus, it is representative of two types of policy evaluation estimators that often give strong empirical performance Voloshin et al. [2019].

Our objective is to determine the sequence of behavior policies that minimize error in estimation of $v(\pi)$. Formally, we seek to minimize mean squared error which is defined as: $\mathbb{E}_\mathcal{D}\left[(Y_n - v(\pi))^2\right]$ where the expectation is over the collected data set $\mathcal{D}$.

### 2.3 RELATED WORK

Our paper builds upon work in the bandit literature for optimal data collection for estimating a weighted sum of the mean reward associated with each arm. Antos et al. [2008] study estimating the mean reward of each arm equally well and show that the optimal solution is to pull each arm proportional to the variance of its reward distribution. Since the variances are unknown a priori, they introduce an algorithm that pulls arms in proportion to the empirical variance of each reward distribution. Carpentier et al. [2015] extend this work by introducing a weighting on each arm that is equivalent to the target policy action probabilities in our work. They show that the optimal solution is then to pull each arm proportional to the product of the standard deviation of the reward distribution and the arm weighting. Instead of using the empirical standard deviations, they introduce an upper confidence bound on the standard deviation and use it to select actions. Our work is different from these earlier works in that we consider more general tree-structured MDPs of which bandits are a special case.

In RL and MDPs, exploration is widely studied with the objective of finding the optimal policy. Prior work attempts to balance exploration to reduce uncertainty with exploitation to converge to the optimal policy. Common approaches are based on reducing uncertainty [Osband et al., 2016,

O'Donoghue et al., 2018] or incentivizing visitation of novel states [Barto, 2013, Pathak et al., 2017, Burda et al., 2018]. These works differ from our work in that we focus on evaluating a fixed policy rather than finding the optimal policy. In our problem, the trade-off becomes balancing taking actions to reduce uncertainty with taking actions that the target policy is likely to take.

Our work is similar in spirit to work on adaptive importance sampling [Rubinstein and Kroese, 2013] which aims to lower the variance of Monte Carlo estimators by adapting the data collection distribution. Adaptive importance sampling was used by Hanna et al. [2017] to lower the variance of policy evaluation in MDPs. It has also been used to lower the variance of policy gradient RL algorithms [Bouchard et al., 2016, Ciosek and Whiteson, 2017]. AIS methods attempt to find a single optimal sampling distribution whereas our approach attempts to reduce uncertainty in the estimated mean rewards. In a similar spirit, Talebi and Maillard [2019] adapt the behavior policy to minimize error in estimating the transition model $P$.

## 3 OPTIMAL DATA COLLECTION IN MULTI-ARMED BANDITS

Before we address optimal data collection for policy evaluation in MDPs, we first revisit the problem in the bandit setting as addressed by earlier work Carpentier et al. [2015]. The bandit setting provides intuition for how a good data collection strategy should select actions, though it falls short of an entire solution for MDPs.

Observe that the policy value in a bandit problem is defined as $v(\pi) := \sum_{a=1}^{A} \pi(a)\mu(a)$ where the bandit consist of a single state $s$ and $A$ actions indexed as $a = 1, 2, \ldots, A$. In this setting, the horizon $L = 1$ so we return to the same state after taking an action $a$ at time $t$. Hence, we drop the state $s$ from our standard notation.

Suppose we have a budget of $n$ samples to divide between the arms and let $T_n(1), T_n(2), \ldots, T_n(A)$ be the number of samples allocated to actions $1, 2, \ldots, A$ at the end of $n$ rounds. We define the estimate:

$$Y_n := \sum_{a=1}^{A} \frac{\pi(a)}{T_n(a)} \sum_{h=1}^{T_n(a)} R_h(a) = \sum_{a=1}^{A} \pi(a)\widehat{\mu}(a). \quad (1)$$

where, $R_h(a)$ is the $h^{\text{th}}$ reward received after taking action $a$. Note that, once all actions where $\pi(a) > 0$ have been tried, $Y_n$ is an unbiased estimator of $v(\pi)$ since $\widehat{\mu}(a)$ is an unbiased estimator of $\mu(a)$. Thus, reducing MSE requires allocating the $n$ samples to reduce variance. As shown by Carpentier et al. [2015], the minimal-variance allocation is given by pulling each arm with the proportion $b^\star(a) \propto \pi(a)\sigma(a)$. Though this result was previously shown, we prove it for completeness in Proposition 1 in Appendix A. Intuitively,

there is more uncertainty about the mean reward for actions with higher variance reward distributions. Selecting these actions more often is needed to offset higher variance. The optimal proportion also takes $\pi$ into account as a high variance mean reward estimate for one action can be acceptable if $\pi$ would rarely take that action.

Note that sampling according to eq. (13) introduces unnecessary variance compared to deterministically selecting actions to match the optimal proportion. Since the variances are typically unknown, a number of works in the bandit community propose different approaches to estimate the variances for both basic bandits and several related extensions [Antos et al., 2008, Carpentier and Munos, 2011, 2012, Carpentier et al., 2015, Neufeld et al., 2014]. Finally, note that incorporating variance aware techniques has been studied in multi-armed bandits [Audibert et al., 2009, Mukherjee et al., 2018]. However, these works tend to focus on regret minimization, whereas we focus on MSE reduction. However, none of of these works address the fundamental challenge that MDPs bring – action selection must account for both immediate variance reduction in the current state as well as variance reduction in future states visited. In the next section, we begin to address this challenge by deriving minimal-variance action proportions for tree-structured MDPs.

## 4 OPTIMAL DATA COLLECTION IN TREE MDPS

In this section, we derive the optimal action proportions for tree-structured MDPs assuming the variances of the reward distributions are known, introduce an algorithm that approximates the optimal allocation when the variances are unknown, and bound the finite-sample MSE of this algorithm. Tree MDPs are a straightforward extension of the multi-armed bandit model to capture the fact that the optimal allocation for each action in a given state must consider the future states that could arise from taking that action.

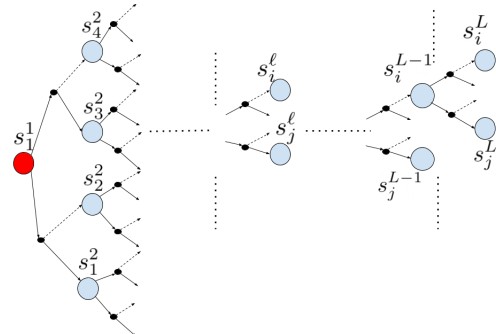

Figure 1: An $L$-depth tree with 2 actions at each state.

We first define a discrete tree MDP as follows:

**Definition 1. (Tree MDP)** An MDP is a discrete tree MDP $\mathbf{T} \subset \mathbf{M}$ (see Figure 1) if the following holds:

**(1)** There are $L$ levels indexed by $\ell$ where $\ell = 1, 2, \ldots, L$.

**(2)** Every state is represented as $s_i^\ell$ where $\ell$ is the level of the state $s$ indexed by $i$.

**(3)** The transition probabilities are such that one can only transition from a state in level $\ell$ to one in level $\ell+1$ and each non-initial state can only be reached through one other state and only one action in that state. Formally, $\forall s', P(s'|s, a) \neq 0$ for only one state-action pair $s, a$ and if $s'$ is in level $\ell+1$ then $s$ is in level $\ell$. Finally, $P(s_j^{L+1}|s_i^L, a) = 0, \forall a$.

**(4)** For simplicity, we assume that there is a single starting state $s_1^1$ (called the root). It is easy to extend our results to multiple starting states with a starting state distribution, $d_0$, by assuming that there is only one action available in the root that leads to each possible start state, $s$, with probability $d_0(s)$. The leaf states are denoted as $s_i^L$.

**(5)** The interaction stops after $L$ steps in state $s_i^L$ after taking an action $a$ and observing the reward $R_L(s_i^L, a)$.

Note that, because we assume a single initial state, $s_1^1$, we have that estimating $v(\pi)$ is equivalent to estimating $v(s_1^1)$. A similar Tree MDP model has been previously used in theoretical analysis by Jiang and Li [2016]; our model is slightly more general as we consider per-step stochastic rewards whereas Jiang and Li [2016] only consider deterministic rewards at the end of trajectories.

### 4.1 ORACLE DATA COLLECTION

We first consider an oracle data collection strategy which knows the variance of all reward distributions and knows the state transition probabilities. After observing $n$ state-action-reward tuples, the oracle computes the following estimate of $v^\pi(s_1^1)$ (or equivalently $v(\pi)$):

$$
\begin{aligned}
Y_n(s_1^1) &:= \sum_{a=1}^A \pi(a|s_1^1)\left(\frac{1}{T_n(s_1^1, a)}\sum_{h=1}^{T_n(s_1^1, a)} R_h(s_1^1, a)\right.\\
&\quad \left. + \gamma \sum_{s_j^{\ell+1}} P(s_j^{\ell+1}|s_1^1, a) Y_n(s_j^2)\right)\\
&= \sum_{a=1}^A \pi(a|s_1^1)\left(\widehat{\mu}(s_1^1, a) + \gamma\sum_{s_j^{\ell+1}} P(s_j^{\ell+1}|s_1^1, a) Y_n(s_j^2)\right) \quad (2)
\end{aligned}
$$

where $T_n(s, a)$ denotes the number of times that the oracle took action $a$ in state $s$. Note that in Section 2 we define $Y_n(s, t)$ but now we use $Y_n(s)$ as timestep is implicit in the layer of the tree. Also (2) differs from the estimator defined in Section 2.2 as it uses the true transition probabilities, $P$, instead of their empirical estimate, $\widehat{P}$. The MSE of $Y_n$ is:

$$
\begin{aligned}
&\mathbb{E}_{\mathcal{D}}[(Y_n(s_1^1) - v^\pi(s_1^1))^2]\\
&= \mathbf{Var}(Y_n(s_1^1)) + \text{bias}^2(Y_n(s_1^1)). \quad (3)
\end{aligned}
$$

The bias of this estimator becomes zero once all $(s, a)$-pairs with $\pi(a|s) > 0$ have been visited a single time, thus we focus on reducing $\mathbf{Var}(Y_n(s_1^1))$. Before defining the oracle data collection strategy, we first state an assumption on $\mathcal{D}$.

**Assumption 1.** *The data $\mathcal{D}$ collected over $n$ state-action-reward samples has at least one observation of each state-action pair, $(s, a)$, for which $\pi(a|s) > 0$.*

Assumption 1 ensures that $Y_n$ is an unbiased estimator of $v(\pi)$ so that reducing MSE is equivalent to reducing variance. Before stating our main result, we provide intuition with a lemma that gives the optimal proportion for each action in a 2-depth tree.

**Lemma 1.** *Let $\mathbf{T}$ be a 2-depth stochastic tree MDP as defined in Definition 1 (see Figure 3 in Appendix B). Let $Y_n(s_1^1)$ be the estimated return of the starting state $s_1^1$ after observing $n$ state-action-reward samples. Note that $v^\pi(s_1^1)$ is the expectation of $Y_n(s_1^1)$ under Assumption 1. Let $\mathcal{D}$ be the observed data over $n$ state-action-reward samples. Minimal MSE, $\mathbb{E}_{\mathcal{D}}[(Y_n(s_1^1) - v^\pi(s_1^1))^2]$, is obtained by taking actions in each state in the following proportions:*

$$
b^*(a|s_j^2) \propto \pi(a|s_j^2)\sigma(s_j^2, a)
$$

$$
b^*(a|s_1^1) \propto \sqrt{\pi^2(a|s_1^1)\left[\sigma^2(s_1^1, a) + \gamma^2 \sum_{s_j^2} P(s_j^2|s_1^1, a)B^2(s_j^2)\right]},
$$

*where, $B(s_j^2) = \sum_a \pi(a|s_j^2)\sigma(s_j^2, a)$.*

**Proof (Overview):** We decompose the MSE into its variance and bias terms and show that $Y_n$ is unbiased under Assumption 1. Next note that the reward in the next state is conditionally independent of the reward in the current state given the current state and action. Hence we can write the variance in terms of the variance of the estimate in the initial state and the variance of the estimate in the final layer. We then rewrite the total samples of a state-action pair i.e $T_n(s_i^\ell, a)$ in terms of the proportion of the number of times the action was sampled in the state i.e $b(a|s_i^\ell)$. To do so, we take into account the tree structure to derive the expected proportion of times that action $a$ is taken in each state in layer 2 as follows:

$$
b(a|s_i^2) = \frac{T_n(s_i^2, a)}{\sum_{a'} T_n(s_i^2, a')} \overset{(a)}{=} \frac{T_n(s_i^2, a)/n}{P(s_i^2|s_1^1, a)T_n(s_1^1, a)/n}
$$

where in $(a)$ the action $a$ is used to transition to state $s_j^2$ from $s_1^1$ and so $\sum_a T_n(s_i^2, a) = P(s_i^2|s_1^1, a)T_n(s_1^1, a)$. We next substitute the $b(a|s_i^\ell)$ for each state-action pair into the variance expression and determine the $b$ values that minimize the expression subject to $\forall s, \sum_a b(a|s) = 1$ and $\forall s, b(a|s) > 0$. The full proof is given in Appendix B. ∎

Note that the optimal proportion in the leaf states, $b^*(a|s_j^2)$, is the same as in Carpentier and Munos [2011] (see Proposition 1) as terminal states can be treated as bandits in which

actions do not affect subsequent states. The key difference is in the root state, $s_1^1$, where the optimal action proportion, $b^*(a|s_1^1)$ depends on the expected leaf state normalization factor $B(s_j^2)$ where $s_j^2$ is a state sampled from $P(\cdot|s_1^1, a)$. The normalization factor, $B(s_i^2)$, captures the total contribution of state $s_i^2$ to the variance of $Y_n$ and thus actions in the root state must be chosen to 1) reduce variance in the immediate reward estimate and to 2) get to states that contribute more to the variance of the estimate. We explore the implications of the oracle action proportions in Lemma 1 with the following two examples.

**Example 1. (Child Variance matters)** Consider a 2-depth, 2-action tree MDP **T** with deterministic $P$, i.e., $P(s_2^2|s_1^1, 2) = P(s_1^2|s_1^1, 1) = 1$ and $\gamma = 1$ (see Figure 4 (Left) in Appendix C). Suppose the target policy is the uniform distribution in all states so that $\forall(s, a), \pi(a|s) = \frac{1}{2}$. The reward distribution variances are given by $\sigma^2(s_1^1, 1) = 400$, $\sigma^2(s_1^1, 2) = 600$, $\sigma^2(s_1^2, 1) = 400$, $\sigma^2(s_1^2, 2) = 400$, $\sigma^2(s_2^2, 1) = 4$, and $\sigma^2(s_2^2, 2) = 4$. So the right sub-tree at $s_1^1$ has higher variance (larger $B$-value) than the left sub-tree. Following the sampling rule in Lemma 1 we can show that $b^*(1|s_1^1) > b^*(2|s_1^1)$ (the full calculation is given in Appendix C). Hence the right sub-tree with higher variance will have a higher proportion of pulls which allows the oracle to get to the high variance $s_1^2$. Observe that treating $s_1^1$ as a bandit leads to choosing action 2 more often as $\sigma^2(s_1^1, 2) > \sigma^2(s_1^1, 1)$. However, taking action 2 leads to state $s_2^2$ which contributes much less to the total variance. Thus, this example highlights the need to consider the variance of subsequent states.

**Example 2. (Transition Model matters)** Consider a 2-depth, 2-action tree MDP **T** in which we have $P(s_1^2|s_1^1, 1) = p$, $P(s_2^2|s_1^1, 1) = 1 - p$, $P(s_3^2|s_1^1, 2) = p$, and $P(s_4^2|s_1^1, 2) = 1 - p$. This example is shown in Figure 4 (Right) in Appendix C. Following the result of Lemma 1 if $p \gg (1 - p)$ it can be shown that the variances of the states $s_1^2$ and $s_3^2$ have greater importance in calculating the optimal sampling proportions of $s_1^1$. The calculation is shown in Appendix D. Thus, less likely future states have less importance for computing the optimal sampling proportion in a given state.

Having developed intuition for minimal-variance action selection in a 2-depth tree MDP, we now give our main result that extends Lemma 1 to an $L$-depth tree.

**Theorem 1.** *Assume the underlying MDP is an $L$-depth tree MDP as defined in Definition 1. Let the estimated return of the starting state $s_1^1$ after $n$ state-action-reward samples be defined as $Y_n(s_1^1)$. Note that the $v^\pi(s_1^1)$ is the expectation of $Y_n(s_1^1)$ under Assumption 1. Let $\mathcal{D}$ be the observed data over $n$ state-action-reward samples. To minimize MSE $\mathbb{E}_\mathcal{D}[(Y_n(s_1^1)) - \mu(Y_n(s_1^1)))^2]$ the optimal sampling propor-*

*tions for any arbitrary state is given by:*

$$b^*(a|s_i^\ell) \propto \sqrt{\pi^2(a|s_i^\ell)\left[\sigma^2(s_i^\ell, a) + \gamma^2 \sum_{s_j^{\ell+1}} P(s_j^{\ell+1}|s_i^\ell, a)B^2(s_j^{\ell+1})\right]},$$

*where, $B(s_j^2)$ is the normalization factor defined as follows:*

$$B(s_i^\ell) = \sum_a \sqrt{\pi^2(a|s_i^\ell)\left(\sigma^2(s_i^\ell, a) + \gamma^2 \sum_{s_j^{\ell+1}} P(s_j^{\ell+1}|s_i^\ell, a)B^2(s_j^{\ell+1})\right)} \quad (4)$$

**Proof (Overview):** We prove Theorem 1 by induction. Lemma 1 proves the base case of estimating the sampling proportion for level $L - 1$ and $L$. Then, for the induction step, we assume that all the sampling proportions from level $L$ till some arbitrary level $\ell + 1$ can be subsequently built up using dynamic programming starting from level $L$. For states in level $L$ to the states in level $\ell + 1$ we can compute $b^*(a|s_i^{\ell+1})$ by repeatedly applying Lemma 1. Then we show that at the level $\ell$ we get a similar recursive sampling proportion as stated in the theorem statement. The proof is given in Appendix E. ∎

### 4.2 MSE OF THE ORACLE

In this subsection, we derive the MSE that the oracle will incur when matching the action proportions given by Theorem 1. The oracle is run for $K$ episodes where each episode consist of $L$ length trajectory of visiting state-action pairs. So the total budget is $n = KL$. At the end of the $K$-th episode the MSE of the oracle is estimated which is shown in Proposition 2. Before stating the proposition we introduce additional notation which we will use throughout the remainder of the paper. Let

$$T_t^k(s, a) = \sum_{i=0}^{k-1} \mathbb{I}\left\{\left(s_t^i, a_t^i\right) = (s, a)\right\}, \forall t, s, a \quad (5)$$

denote the total number of times that $(s, a)$ has been observed in $\mathcal{D}$ (across all trajectories) up to time $t$ in episode $k$ and $\mathbb{I}\{\cdot\}$ is the indicator function. Similarly let

$$T_t^k(s, a, s') = \sum_{i=0}^{k-1} \mathbb{I}\{\left(s_t^i, a_t^i, s_{t+1}^i\right) = (s, a, s')\}, \forall t, s, a, s' \quad (6)$$

denote the number of times action $a$ is taken in $s$ to transition to $s'$. Finally we define the state sample $T_t^k(s) = \sum_a T_t^k(s, a)$ as the total number of times any state is visited and an action is taken in that state.

**Proposition 2.** *Let there be an oracle which knows the state-action variances and transition probabilities of the $L$-depth tree MDP **T**. Let the oracle take actions in the proportions given by Theorem 1. Let $\mathcal{D}$ be the observed data over $n$*

*state-action-reward samples such that $n = KL$. Then the oracle suffers an MSE of*

$$\mathcal{L}_n^* = \sum_{\ell=1}^{L} \left[ \frac{B^2(s_i^\ell)}{T_L^{*,K}(s_i^\ell)} \right.$$
$$\left. + \gamma^2 \sum_a \pi^2(a|s_i^\ell) \sum_{s_j^{\ell+1}} P(s_j^{\ell+1}|s_i^\ell, a) \frac{B^2(s_j^{\ell+1})}{T_L^{*,K}(s_j^{\ell+1})} \right]. \quad (7)$$

*where, $T_L^{*,K}(s_i^\ell)$ denotes the optimal state samples of the oracle at the end of episode $K$.*

The proof is given in Appendix F. From Proposition 2 we see that the MSE of the oracle goes to 0 as the number of episodes $K \to \infty$, and $T_L^{*,K}(s_i^\ell) \to \infty$ simultaneously for all $s_i^\ell \in \mathcal{S}$. Observe that if for every state $s$ the total state counts $T_L^{*,K}(s) = cn$ for some constant $c > 0$ then the loss of the oracle goes to 0 at the rate $O(1/n)$.

## 4.3 REDUCED VARIANCE SAMPLING

The oracle data collection strategy provides intuition for optimal data collection for minimal-variance policy evaluation, however, it is *not* a practical strategy itself as it requires $\sigma$ and $P$ to be known. We now introduce a practical data collection algorithm – **Re**duced **Var**iance Sampling (ReVar) – that is agnostic to $\sigma$ and $P$. Our algorithm follows the proportions given by Theorem 1 with the true reward variances replaced with an upper confidence bound and the true transition probabilities replaced with empirical frequencies. Formally, we define the desired proportion for action $a$ in state $s_i^\ell$ after $t$ steps as $\widehat{b}_{t+1}^k(a|s_i^\ell) \propto$

$$\sqrt{\pi^2(a|s_i^\ell) \left[ \widehat{\sigma u}_t^{(2),k}(s_i^\ell, a) + \gamma^2 \sum_{s_j^{\ell+1}} \widehat{P}_t^k(s_j^{\ell+1}|s_i^\ell, a) \widehat{B}_t^{(2),k}(s_j^{\ell+1}) \right]}, \quad (8)$$

The upper confidence bound on the variance $\sigma^2(s_i^\ell, a)$, denoted by $\widehat{\sigma u}_{t-1}^{(2),k}(s_i^\ell, a) = (\widehat{\sigma u}_t^k(s_i^\ell, a))^2$, is defined as:

$$\widehat{\sigma u}_t^k(s_i^\ell, a) := \widehat{\sigma}_t^k(s_i^\ell, a) + 2c\sqrt{\frac{\log(SAn(n+1)/\delta)}{T_t^k(s_i^\ell, a)}} \quad (9)$$

where, $\widehat{\sigma}_t^k(s_i^\ell, a)$ is the plug-in estimate of the standard deviation $\sigma(s_i^\ell, a)$, $c > 0$ is a constant depending on the boundedness of the rewards to be made explicit later, and $n = KL$ is the total budget of samples. Using an upper confidence bound on the reward standard deviations captures our uncertainty about $\sigma(s_i^\ell, a)$ needed to compute the true optimal proportions. The state transition model is estimated as:

$$\widehat{P}_t^k(s_j^{\ell+1}|s_i^\ell, a) = \frac{T_t^k(s_i^\ell, a, s_j^{\ell+1})}{T_t^k(s_i^\ell, a)} \quad (10)$$

where, $T_t^k(s_i^\ell, a, s_j^{\ell+1})$ is defined in (6). Further in (8), $\widehat{B}_t^k(s_j^{\ell+1})$ is the plug-in estimate of $B(s_j^{\ell+1})$. Observe that for all of these plug-in estimates we use all the past history till time $t$ in episode $k$ to estimate these statistics.

Eq. (8) allows us to estimate the optimal proportion for all actions in any state. To match these proportions, rather than sampling from $\widehat{b}_{t+1}^k(a|s_i^\ell)$, ReVar takes action $I_{t+1}^k$ at time $t + 1$ in episode $k$ according to:

$$I_{t+1}^k = \arg\max_a \left\{ \frac{\widehat{b}_t^k(a|s_i^\ell)}{T_t^k(s_i^\ell, a)} \right\}. \quad (11)$$

This action selection rule ensures that the ratio $\widehat{b}_t^k(a|s_i^\ell)/T_t(s_i^L, a) \approx 1$. It is a deterministic action selection rule and thus avoids variance due to simply sampling from the estimated optimal proportions. Note that in the terminal states, $s_i^L$, the sampling rule becomes

$$I_{t+1}^k = \arg\max_a \left\{ \frac{\pi(a|s_i^L)\widehat{\sigma u}_t^k(s_i^L, a)}{T_t^k(s_i^L, a)} \right\}$$

which matches the bandit sampling rule of Carpentier and Munos [2011, 2012].

We give pseudocode for ReVar in Algorithm 1. The algorithm proceeds in episodes. In each episode we generate a trajectory from the starting state $s_1^1$ (root) to one of the terminal state $s_j^L$ (leaf). At episode $k$ and time-step $t$ in some arbitrary state $s_i^\ell$ the next action $I_{t+1}$ is chosen based on (11). The trajectory generated is added to the dataset $\mathcal{D}$. At the end of the episode we update the model parameters, i.e. we estimate the $\widehat{\sigma}_t^k(s_i^\ell, a)$, and $\widehat{P}_t^k(s_i^{\ell+1}|s_j^\ell, a)$ for each state-action pair. Finally, we update $\widehat{b}_1^{k+1}(a|s_\ell^i)$ for the next episode using eq. (9).

---

**Algorithm 1 Re**duced **Vari**ance Sampling (ReVar )

---

1: **Input:** Number of trajectories to collect, $K$.
2: **Output:** Dataset $\mathcal{D}$.
3: Initialize $\mathcal{D} = \emptyset$, $\widehat{b}_1^0(a|s_i^\ell)$ uniform over all actions in each state.
4: **for** $k \in 0, 1, \ldots, K$ **do**
5:     Generate trajectory $H^k := \{S_t, I_t, R(I_t)\}_{t=1}^L$ by selecting $I_t$ according to (11).
6:     $\mathcal{D} \leftarrow \mathcal{D} \cup \{(H^k, \widehat{b}_L^k)\}$
7:     Update model parameters and estimate $\widehat{b}_1^{k+1}(a|s_i^\ell)$ for each $(s_i^\ell, a)$.
8:     Update $\widehat{b}_1^{k+1}(a|s_i^\ell)$ from level $L$ to 1 following (8).
9: **Return** Dataset $\mathcal{D}$ to evaluate policy $\pi$.

---

## 4.4 REGRET ANALYSIS

We now theoretically analyze ReVar by bounding its regret with respect to the oracle behavior policy. We analyze ReVar

under the assumption that $P$ is known and so we are only concerned with obtaining accurate estimates of the reward means and variances. This assumption is only made for the regret analysis and is *not* a fundamental requirement of ReVar. Though somewhat restrictive, the case of known state transitions is still interesting as it arises in practice when state transitions are deterministic or we can estimate $P$ much easier than we can estimate the reward means.

We first define the notion of regret of an algorithm compared to the oracle MSE $\mathcal{L}_n^*$ in (7) as follows:

$$\mathcal{R}_n = \mathcal{L}_n - \mathcal{L}_n^*$$

where, $n$ is the total budget, and $\mathcal{L}_n$ is the MSE at the end of episode $K$ following the sampling rule in (8). We make the following assumption that rewards are bounded:

**Assumption 2.** *The reward from any state-action pair has bounded range, i.e., $R_t(s,a) \in [-\eta, \eta]$ almost surely at every time-step $t$ for some fixed $\eta > 0$.*

Note that this is a common assumption in the RL literature [Munos, 2005, Agarwal et al., 2019]. The reward can also be multi-modal as long as it is bounded. Then the regret of ReVar over a $L$-depth deterministic tree is given by the following theorem.

**Theorem 2.** *Let the total budget be $n = KL$ and $n \geq 4SA$. Then the total regret in a deterministic $L$-depth $\mathbf{T}$ at the end of $K$-th episode when taking actions according to (8) is given by*

$$\mathcal{R}_n \leq \widetilde{O}\left(\frac{B_{s_1^1}^2 \sqrt{\log(SAn^{11/2})}}{n^{3/2} b_{\min}^{*,3/2}(s_1^1)}\right.$$
$$\left. + \gamma \sum_{\ell=2}^{L} \max_{s_j^\ell, a} \pi(a|s_1^1) P(s_j^\ell|s_1^1, a) \frac{B_{s_j^\ell}^2 \sqrt{\log(SAn^{11/2})}}{n^{3/2} b_{\min}^{*,3/2}(s_j^\ell)}\right)$$

*where, the $\widetilde{O}$ hides other lower order terms and $B_{s_i^\ell}$ is defined in (4) and $b_{\min}^*(s) = \min_a b^*(a|s)$.*

Note that if $L = 1$, $|\mathcal{S}| = 1$, we recover the bandit setting and our regret bound matches the bound in Carpentier and Munos [2011]. Note that MSE using data generated by any policy decays at a rate no faster than $O(n^{-1})$, the parametric rate. The key feature of ReVar is that it converges to the oracle policy. This means that asymptotically, the MSE based on ReVar will match that of the oracle. Theorem 2 shows that the regret scales like $O(n^{-3/2})$ if we have the $b_{\min}^*(s)$ over all states $s \in \mathcal{S}$ as some reasonable constant $O(1)$. In contrast, suppose we sample trajectories from a suboptimal policy, i.e., a policy that produces an MSE worse than that of the oracle for every $n$. This MSE gap never diminishes, so the regret cannot decrease at a rate faster than the oracle rate of $O(n^{-1})$. Finally, note that the regret

bound in Theorem 2 is a problem dependent bound as it involves the parameter $b_{\min}^*(s)$.

**Proof (Overview):** We decompose the proof into several steps. We define the good event $\xi_\delta$ based on the state-action-reward samples $\mathcal{D}$ that holds for all episode $k$ and time $t$ such that $|\hat{\sigma}_t^k(s,a) - \sigma(s,a)| \leq \epsilon$ for some $\epsilon > 0$ with probability $1 - \delta$ made explicit in Corollary 1 . Now observe that MSE of ReVar is

$$\mathcal{L}_n = \mathbb{E}_\mathcal{D}\left[\left(Y_n(s_1^1)) - v^\pi(s_1^1)\right)^2\right]$$
$$= \mathbb{E}_\mathcal{D}\left[\left(Y_n(s_1^1)) - v^\pi(s_1^1)\right)^2 \mathbb{I}\{\xi_\delta\}\right]$$
$$+ \mathbb{E}_\mathcal{D}\left[\left(Y_n(s_1^1)) - v^\pi(s_1^1)\right)^2 \mathbb{I}\{\xi_\delta^C\}\right] \quad (12)$$

Note that here we are considering a known transition function $P$. The first term in (12) can be bounded using

$$\mathbb{E}_\mathcal{D}\left[\left(Y_n(s_1^1)) - v^\pi(s_1^1)\right)^2 \mathbb{I}\{\xi_\delta\}\right] = \mathbf{Var}[Y_n(s_1^1)]\mathbb{E}[T_n^k(s_1^1)]$$
$$\leq \sum_a \pi^2(a|s_1^1)\left[\frac{\sigma^2(s_1^1, a)}{\underline{T}_n^{(2),k}(s_1^1, a)}\right]\mathbb{E}[T_n^k(s_1^1, a)]$$
$$+ \gamma^2 \sum_a \pi^2(a|s_1^1) \sum_{s_j^2} P^2(s_j^2|s_1^1, a)$$
$$\cdot \sum_{a'} \pi^2(a'|s_j^2)\left[\frac{\sigma^2(s_j^2, a')}{\underline{T}_n^{(2),k}(s_j^2, a')}\right]\mathbb{E}[T_n^k(s_j^2, a')]$$

where, $\underline{T}^{(2),k}(s_1^1, a)$ is a lower bound to $T^{(2),k}(s_1^1, a)$ made explicit in Lemma 7, and $\underline{T}^{(2),k}(s_j^2, a)$ is a lower bound to $T^{(2),k}(s_1^1, a)$ made explicit in Lemma 6. We can combine these two lower bounds and give an upper bound to MSE in a two depth $\mathbf{T}$ which is shown Lemma 8. Finally, for the $L$ depth stochastic tree we can repeatedly apply Lemma 8 to bound the first term. For the second term we set the $\delta = n^{-2}$ and use the boundedness assumption in Assumption 2 to get the final bound. The proof is given in Appendix H. ∎

## 5  OPTIMAL DATA COLLECTION BEYOND TREES

The tree-MDP model considered above allows us to develop a foundation for minimal-variance data collection in decision problems where actions at one state affect subsequent states. One limitation of this model is that, for any non-initial state, $s_i^\ell$, there is only a single state-action path that could have been taken to reach it. In a more general finite-horizon MDP, there could be many different paths to reach the same non-initial state. Unfortunately, the existence of multiple paths to a state introduces cyclical dependencies between states that complicate derivation of the minimal-variance data collection strategy and regret analysis. In this section, we elucidate this difficulty by considering the class of directed acyclic graph (DAG) MDPs.

In this section we first define a DAG $\mathcal{G} \subset \mathbf{M}$. An illustrative figure of a 3-depth 2-action $\mathcal{G}$ is in Figure 5 of Appendix I .

**Definition 2. (DAG MDP)** A DAG MDP follows the same definition as the tree MDP in Definition 1 except $P(s'|s,a)$ can be non-zero for any $s$ in layer $\ell$, $s'$ in layer $\ell+1$, and any $a$, i.e., one can now reach $s'$ through multiple previous state-action pairs.

**Proposition 3.** *Let $\mathcal{G}$ be a $3$-depth, $A$-action DAG defined in Definition 2. The minimal-MSE sampling proportions $b^*(a|s_1^1), b^*(a|s_j^2)$ depend on themselves such that $b(a|s_1^1) \propto f(1/b(a|s_1^1))$ and $b(a|s_j^2) \propto f(1/b(a|s_j^2))$ where $f(\cdot)$ is a function that hides other dependencies on variances of $s$ and its children.*

The proof technique follows the approach of Lemma 1 but takes into account the multiple paths leading to the same state. The possibility of multiple paths results in the cyclical dependency of the sampling proportions in level 1 and 2. Note that in $\mathbf{T}$ there is a single path to each state and this cyclical dependency does not arise. The full proof is given in Appendix I. Because of this cyclical dependency it is difficult to estimate the optimal sampling proportions in $\mathcal{G}$. However, we can approximate the optimal sampling proportion that ignores the multiple path problem in $\mathcal{G}$ by using the tree formulation in the following way: At every time $t$ during a trajectory $\tau^k$ call the Algorithm 2 in Appendix J to estimate $B_0(s)$ where $B_{t'}(s) \in \mathbb{R}^{L \times |\mathcal{S}|}$ stores the expected standard deviation of the state $s$ at iteration $t'$. After $L$ such iteration we use the value $B_0(s)$ to estimate $b(a|s)$ as follows:

$$b^*(a|s) \propto \sqrt{\pi^2(a|s)\Big[\sigma^2(s,a)+\gamma^2\sum_{s'}P(s'|s,a)B_0^2(s)\Big]}.$$

Note that for a terminal state $s$ we have the transition probability $P(s'|s,a) = 0$ and then the $b(a|s) = \pi(a|s)\sigma(s,a)$. This iterative procedure follows from the tree formulation in Theorem 1 and is necessary in $\mathcal{G}$ to take into account the multiple paths to a particular state. Also observe that in Algorithm 2 we use value-iteration for the episodic setting [Sutton and Barto, 2018] to estimate the the optimal sampling proportion iteratively.

# 6  EMPIRICAL STUDY

We next verify our theoretical findings with simulated policy evaluation tasks in both a tree MDP and a non-tree GridWorld domain. Our experiments are designed to answer the following questions: 1) can ReVar produce policy value estimates with MSE comparable to the oracle solution? and 2) does our novel algorithm lower MSE relative to on-policy sampling of actions? Full implementation details are given in Appendix J.

**Experiment 1 (Tree):** In this setting we have a $4$-depth 2-action deterministic tree MDP $\mathbf{T}$ consisting of 15 states. Each state has a low variance arm with $\sigma^2(s,1) = 0.01$ and high target probability $\pi(1|s) = 0.95$ and a high variance arm with $\sigma^2(s,1) = 20.0$ and low target probability

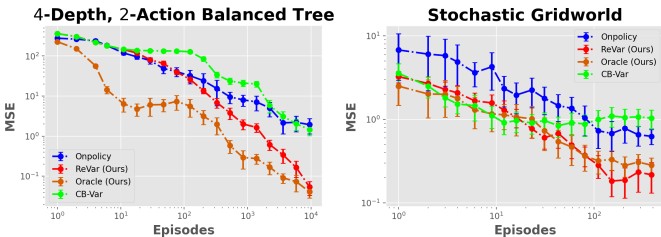

Figure 2: (Left) Deterministic $4$-depth Tree. (Right) Stochastic gridworld. The vertical axis gives MSE and the horizontal axis is the number of episodes collected. Axes use a log-scale and confidence bars show one standard error.

$\pi(2|s) = 0.05$. Hence, the Onpolicy sampling which samples according to $\pi$ will sample the second (high variance) arm less and suffer a high MSE. The CB-Var policy is a bandit policy that uses an empirical Bernstein Inequality [Maurer and Pontil, 2009] to sample an action without looking ahead and suffers high MSE. The Oracle has access to the model and variances and performs the best. ReVar lowers MSE comparable to Onpolicy and CB-Var and eventually matches the oracle's MSE.

**Experiment 2 (Gridworld):** In this setting we have a $4 \times 4$ stochastic gridworld consisting of 16 grid cells. Considering the current episode time-step as part of the state, this MDP is a DAG MDP in which there are multiple path to a single state. There is a single starting location at the top-left corner and a single terminal state at the bottom-right corner. Let $\mathbf{L}, \mathbf{R}, \mathbf{D}, \mathbf{U}$ denote the left, right, down and up actions in every state. Then in each state the right and down actions have low variance arms with $\sigma^2(s,\mathbf{R}) = \sigma^2(s,\mathbf{D}) = 0.01$ and high target policy probability $\pi(\mathbf{R}|s) = \pi(\mathbf{D}|s) = 0.45$. The left and top actions have high variance arms with $\sigma^2(s,\mathbf{L}) = \sigma^2(s,\mathbf{U}) = 0.01$ and low target policy probability $\pi(\mathbf{L}|s) = \pi(\mathbf{U}|s) = 0.05$. Hence, Onpolicy which goes right and down with high probability (to reach the terminal state) will sample the low variance arms more and suffer a high MSE. Similar to above, CB-Var fails to look ahead when selecting actions and thus suffers from high MSE. ReVar lowers MSE compared to Onpolicy and CB-Var and actually matches and then reduces MSE compared to the Oracle. We point out that the DAG structure of the Gridworld violates the tree-structure under which Oracle and ReVar were derived. Nevertheless, both methods lower MSE compared to Onpolicy.

# 7  CONCLUSION AND FUTURE WORKS

This paper has studied the question of how to take actions for minimal-variance policy evaluation of a fixed target policy. We developed a theoretical foundation for data collection in policy evaluation by deriving an oracle data collection policy for the class of finite, tree-structured MDPs. We then introduced a practical algorithm, ReVar, that approximates the oracle strategy by computing an upper confidence bound

on the variance of the future cumulative reward at each state and using this bound in place of the true variances in the oracle strategy. We bound the finite-sample regret (excess MSE) of our algorithm relative to the oracle strategy. We also present an empirical study where we show that ReVar decreases the MSE of policy evaluation relative to several baseline data collection strategies including on-policy sampling. In the future, we would like to extend our derivation of optimal data collection strategies and regret analysis of ReVar to a more general class of MDPs, in particular, relaxing the tree structure and also considering infinite-horizon MDPs. Finally, real world problems often require function approximation to deal with large state and action spaces. This setting raises new theoretical and implementation challenges for ReVar where we intend to incorporate experimental design approaches [Pukelsheim, 2006, Mason et al., 2021, Mukherjee et al., 2022]. Another interesting direction is to incorporate structure in the reward distribution of arms Gupta et al. [2021, 2020]. Addressing these challenges is an interesting direction for future work.

**Acknowledgements:** The authors will like to thank Kevin Jamieson from Allen School of Computer Science & Engineering, University of Washington for pointing out several useful references. This work was partially supported by AFOSR/AFRL grant FA9550-18-1-0166.

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

## A  OPTIMAL SAMPLING IN BANDIT SETTING

**Proposition 1.** *(Restatement)* *In an $A$-action bandit setting, the estimated return of $\pi$ after $n$ action-reward samples is denoted by $Y_n$ as defined in (1). Note that the expectation of $Y_n$ after each action has been sampled once is given by $v(\pi)$. Minimal MSE, $\mathbb{E}_{\mathcal{D}}\left[(Y_n - v(\pi))^2\right]$, is obtained by taking actions in the proportion:*

$$b^*(a) = \frac{\pi(a)\sigma(a)}{\sum_{a'=1}^{A} \pi(a')\sigma(a')}. \tag{13}$$

*where $b^*(a)$ denotes the optimal sampling proportion.*

*Proof.* Recall that we have a budget of $n$ samples and we are allowed to draw samples from their respective distributions. Suppose we have $T_n(1), T_n(2), \ldots, T_n(A)$ samples from actions $1, 2, \ldots, A$. Then we can calculate the estimator

$$Y_n = \frac{1}{n}\sum_{t=1}^{n} Y_t = \sum_{a=1}^{A} \frac{\pi(a)}{T_n(a)} \sum_{i=1}^{T_n(a)} R_i(a)$$

where, $n = \sum_{a=1}^{A} T_n(a)$ samples and $R_i(a)$ is the $i^{\text{th}}$ reward received after taking action $a$. We collect a dataset $\mathcal{D}$ of $n$ action-reward samples. Now we use the MSE to estimate how close is $Y_n$ to $v(\pi)$ as follows:

$$\mathbb{E}_{\mathcal{D}}\left[(Y_n - v(\pi))^2\right] = \mathbf{Var}(Y_n) + \text{bias}^2(Y_n).$$

Note that once we have sampled each action once, since $\mathbb{E}_{\mathcal{D}}[Y_n] = v(\pi)$ so $\text{bias}(Y_n) = 0$. So we need to focus only on variance. We can decompose the variance as follows:

$$\mathbf{Var}(Y_n) \overset{(a)}{=} \sum_{a=1}^{A} \mathbf{Var}\left(\frac{\pi(a)}{T_n(a)} \sum_{i=1}^{T_n(a)} R_i(a)\right)$$

$$= \sum_{a=1}^{A} \frac{\pi^2(a)}{T_n^2(a)} \sum_{i=1}^{T_n(a)} \mathbf{Var}\left(R_i(a)\right) = \sum_{a=1}^{A} \frac{\pi^2(a)\sigma^2(a)}{T_n(a)}$$

where, $(a)$ follows as $R_i(a)$ and $R_{i'}(a')$ are independent for every $(i, i')$ and $(a, a')$ pairs. Now we want to optimize $T_n(1), T_n(2), \ldots, T_n(A)$ so that the variance $\mathbf{Var}(Y_n)$ is minimized. We can do this as follows: Let's first write the variance in terms of the proportion $\mathbf{b} \coloneqq \{b(1), b(2), \ldots, b(A)\}$ such that

$$b(a) = \frac{T_n(a)}{\sum_{a'=1}^{A} T_n(a')}.$$

We can then rewrite the optimization problem as follows:

$$\min_{\mathbf{b}} \sum_{a=1}^{A} \frac{\pi^2(a)\sigma^2(a)}{b(a)}, \quad \textbf{s.t.} \sum_{a} b(a) = 1$$

$$\forall a, b(a) > 0. \tag{14}$$

Note that we use $b(a)$ to denote the optimization variable and $b^*(a)$ to denote the optimal sampling proportion. Given this optimization in (14) we can get a closed form solution by introducing the Lagrange multiplier as follows:

$$L(\mathbf{b}, \lambda) = \sum_{a=1}^{A} \frac{\pi^2(a)\sigma^2(a)}{b(a)} + \lambda\left(\sum_{a=1}^{A} b(a) - 1\right). \tag{15}$$

Now to get the Karush-Kuhn-Tucker (KKT) condition we differentiate (15) with respect to $b(a)$ and $\lambda$ as follows:

$$\nabla_{b(a)}L(\mathbf{b}, \lambda) = -\frac{\pi^2(a)\sigma^2(a)}{b^2(a)} + \lambda \tag{16}$$

$$\nabla_{\lambda}L(\mathbf{b}, \lambda) = \sum_{a} b(a) - 1. \tag{17}$$

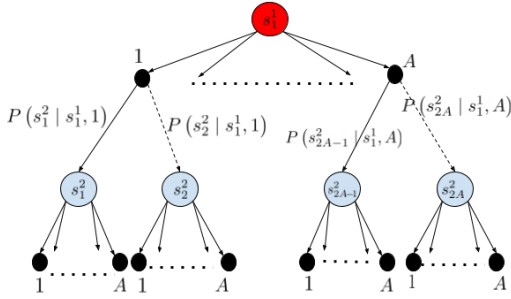

Figure 3: 2-Depth, $A$-action Tree MDP

Now equating (16) and (17) to zero and solving for the solution we obtain:

$$\lambda = \frac{\pi^2(a)\sigma^2(a)}{b^2(a)} \implies b(a) = \sqrt{\frac{\pi^2(a)\sigma^2(a)}{\lambda}}$$

$$\sum_a b(a) = 1 \implies \sum_{a=1}^{A} \sqrt{\frac{\pi^2(a)\sigma^2(a)}{\lambda}} = 1 \implies \sqrt{\lambda} = \sum_{a=1}^{A} \sqrt{\pi^2(a)\sigma^2(a)}.$$

This gives us the optimal sampling proportion

$$b^*(a) = \frac{\pi(a)\sigma(a)}{\sum_{a'=1}^{A} \sqrt{\pi(a')^2\sigma^2(a')}} \implies b^*(a) = \frac{\pi(a)\sigma(a)}{\sum_{a'=1}^{A} \pi(a')\sigma(a')}.$$

Finally, observe that the above optimal sampling for the bandit setting for an action $a$ only depends on the standard deviation $\sigma(a)$ of the action. $\qquad\square$

## B  OPTIMAL SAMPLING IN THREE STATE STOCHASTIC TREE MDP

**Lemma 1.** *(Restatement) Let $\mathbf{T}$ be a 2-depth stochastic tree MDP as defined in Definition 1 (see Figure 3 in Appendix B). Let $Y_n(s_1^1)$ be the estimated return of the starting state $s_1^1$ after observing $n$ state-action-reward samples. Note that $v^\pi(s_1^1)$ is the expectation of $Y_n(s_1^1)$ under Assumption 1. Let $\mathcal{D}$ be the observed data over $n$ state-action-reward samples. To minimise MSE, $\mathbb{E}_{\mathcal{D}}[(Y_n(s_1^1) - v^\pi(s_1^1))^2]$, is obtained by taking actions in each state in the following proportions:*

$$b^*(a|s_j^2) \propto \pi(a|s_j^2)\sigma(s_j^2, a)$$

$$b^*(a|s_1^1) \propto \sqrt{\pi^2(a|s_1^1)\left[\sigma^2(s_1^1, a) + \gamma^2 \sum_{s_j^2} P(s_j^2|s_1^1, a)B^2(s_j^2)\right]},$$

*where, $B(s_j^2) = \sum_a \pi(a|s_j^2)\sigma(s_j^2, a)$.*

*Proof.* We define an estimator $Y_n(s)$ that visits each state-action pair $\sum_{s \in \mathcal{S}} \sum_a T_n(s, a) = n$ times and then plug-ins the estimated sample mean. For the $i^{\text{th}}$ state in level $\ell$ of the tree, this estimator is given as:

$$Y_n(s_i^\ell) = \sum_{a=1}^{A} \left( \underbrace{\frac{\pi(a|s_i^\ell)}{T_n(s_i^\ell, a)} \sum_{h=1}^{T_n(s_i^\ell, a)} R_h(s_i^\ell, a)}_{\text{mean reward weighted by } \pi(a|s_i^\ell)} + \gamma\pi(a|s_i^\ell) \sum_{s_j^{\ell+1}} P(s_j^{\ell+1}|s_i^\ell, a) \underbrace{Y_n(s_j^{\ell+1})}_{\text{Next state estimate}} \right), \text{if } \ell \neq L$$

where we take $Y_n(s_j^{l+1}) = 0$ if $s_i^l$ is a leaf state (i.e., $l = L$).

**Step 1 ($Y_n(s_1^1)$ is an unbiased estimator of $v(\pi)$):** We first show that $Y_n(s_1^1)$ is an unbiased estimator of $v(\pi)$. We use this fact to show that minimizing variance is equivalent to minimizing MSE. The expectation of $Y_n(s_1^1)$ is given as:

$$\mathbb{E}[Y_n(s_1^1)] = \mathbb{E}\left[\sum_{a=1}^{A}\left(\sum_{s_j^2}\frac{\pi(a|s_1^1)}{T_n(s_1^1,a)}\sum_{h=1}^{T_n(s_1^1,a)}R_h(s_1^1,a) + \gamma\pi(a|s_1^1)\sum_{s_j^2}P(s_j^2|s_1^1,a)Y_n(s_j^2)\right)\right]$$

$$\overset{(a)}{=} \sum_{a=1}^{A}\left(\sum_{s_j^2}\frac{\pi(a|s_1^1)}{T_n(s_1^1,a)}\sum_{h=1}^{T_n(s_1^1,a)}\mathbb{E}\left[R_h(s_1^1,a)\right] + \gamma\pi(a|s_1^1)\sum_{s_j^2}P(s_j^2|s_1^1,a)\mathbb{E}\left[Y_n(s_j^2)\right]\right)$$

$$= \sum_{a=1}^{A}\left(\sum_{s_j^2}\frac{\pi(a|s_1^1)}{T_n(s_1^1,a)}T_n(s_1^1,a)\mathbb{E}\left[R_h(s_1^1,a)\right] + \gamma\pi(a|s_1^1)\sum_{s_j^2}P(s_j^2|s_1^1,a)\mathbb{E}\left[Y_n(s_j^2)\right]\right)$$

$$= v^{\pi}(s_1^1)$$

where, $(a)$ follow from the linearity of expectation. Thus, $Y_n(s_1^1)$ is a unbiased estimator of $v(\pi)$.

**Step 2 (Variance of $Y_n(s_1^1)$):** Next we look into the variance of $\mathbf{Var}(Y_n(s_1^1))$.

$$\mathbf{Var}(Y_n(s_1^1)) = \mathbf{Var}\left[\sum_{a=1}^{A}\left(\frac{\pi(a|s_1^1)}{T_n(s_1^1,a)}\sum_{h=1}^{T_n(s_1^1,a)}R_h(s_1^1,a) + \gamma\pi(a|s_1^1)\sum_{s_j^2}P(s_j^2|s_1^1,a)Y_n(s_j^2)\right)\right]$$

$$\overset{(a)}{=} \sum_{a=1}^{A}\left(\frac{\pi^2(a|s_1^1)}{T_n^2(s_1^1,a)}\sum_{h=1}^{T_n(s_1^1,a)}\mathbf{Var}[R_h(s_1^1,a)] + \gamma^2\pi^2(a|s_1^1)\sum_{s_j^2}P^2(s_j^2|s_1^1,a)\mathbf{Var}[Y_n(s_j^2)]\right)$$

$$\overset{(b)}{=} \sum_{a=1}^{A}\left(\frac{\pi^2(a|s_1^1)\sigma^2(s_1^1,a)}{T_n(s_1^1,a)} + \gamma^2\pi^2(a|s_1^1)\sum_{s_j^2}P^2(s_j^2|s_1^1,a)\mathbf{Var}[(Y_n(s_j^2)]\right), \tag{18}$$

where $(a)$ follows because the reward in next state is conditionally independent given the current state and action and $(b)$ follows from $\sigma(s,a) = \mathbf{Var}[R(s,a)]$.

The goal is to reduce the variance $\mathbf{Var}(Y_n(s_1^1))$ in (18). We first unroll the (18) to take into account the conditional behavior probability of each of the path from $s_1^1$ to $s_j^2$ for $j \in \{1,2,3,4\}$. This is shown as follows:

$$\mathbf{Var}(Y_n(s_1^1))) = \sum_{a}\frac{\pi^2(a|s_1^1)\sigma^2(s_1^1,a)}{T_n(s_1^1,a)} + \sum_{a}\sum_{s_j^2}\sum_{a'}\frac{\gamma^2\pi^2(a|s_1^1)P^2(s_j^2|s_1^1,a)\pi^2(a|s_j^2)\sigma^2(s_j^2,a)}{T_n(s_j^2,a)}$$

$$\implies n\mathbf{Var}(Y_n(s_1^1))) = \sum_{a}\frac{\pi^2(a|s_1^1)\sigma^2(s_1^1,a)}{T_n(s_1^1,a)/n} + \sum_{a}\sum_{s_j^2}\sum_{a'}\frac{\gamma^2\pi^2(a|s_1^1)P^2(s_j^2|s_1^1,a)\pi^2(a|s_j^2)\sigma^2(s_j^2,a')}{T_n(s_j^2,a')/n}$$

$$\overset{(a)}{\implies} \sum_{a}\frac{\pi^2(a|s_1^1)\sigma^2(s_1^1,a)}{b(a|s_1^1)} + \sum_{a}\sum_{s_j^2}\sum_{a'}\frac{\gamma^2\pi^2(a|s_1^1)P^2(s_j^2|s_1^1,a)\pi^2(a|s_j^2)\sigma^2(s_j^2,a')}{P(s_j^2|s_1^1,a)b(a|s_1^1)b(a'|s_j^2)}$$

where, $(a)$ follows as

$$b(a'|s_i^2) = \frac{T_n(s_i^2,a')}{\sum_a T_n(s_i^2,a)} \overset{(a)}{=} \frac{T_n(s_i^2,a')}{P(s_i^2|s_1^1,a)T_n(s_1^1,a)} = \frac{T_n(s_i^2,a')/n}{P(s_i^2|s_1^1,a)T_n(s_1^1,a)/n}$$

$$\implies T_n(s_i^2,a')/n = P(s_i^2|s_1^1,a)b(a|s_1^1)b(a'|s_i^2)$$

where, in $(a)$ the action $a$ is used from state $s_1^1$ to transition to state $s_i^2$. Similarly in $(a)$ we can substitute $T_n(s_j^2,a)/n$ for all $s \in \mathcal{S}$. Note that this follows because of the tree MDP structure as path to state $s_j^{\ell+1}$ depends on it immediate parent state $s_i^{\ell}$ (see **(3)** in Definition 1). Recall that $P(s'|s,a)T_n(s'|s,a)$ is the expected times we end up in next state, not the actual number of times. We use $P$ in this formulation instead of $\widehat{P}$ as this is the oracle setting which has access to the transition model and our goal is to minimize the number of samples $n$.

**Step 3 (Minimal Variance Objective function):** Note that we use $b(a|s)$ to denote the optimization variable and $b^*(a|s)$ to denote the optimal sampling proportion. Now, we determine the $b$ values that give minimal variance by minimizing the following objective:

$$\min_{\mathbf{b}} \sum_a \frac{\pi^2(a|s_1^1)\sigma^2(s_1^1,a)}{b(a|s_1^1)} + \sum_a \sum_{s_j^2} \sum_{a'} \frac{\gamma^2\pi^2(a|s_1^1)P(s_j^2|s_1^1,a)\pi^2(a|s_j^2)\sigma^2(s_j^2,a')}{b(a|s_1^1)b(a'|s_j^2)}$$

$$\text{s.t.} \quad \forall s, \quad \sum_a b(a|s) = 1$$

$$\forall s,a \quad b(a|s) > 0. \tag{19}$$

We can get a closed form solution by introducing the Lagrange multiplier as follows:

$$L(\lambda,\mathbf{b}) = \sum_a \frac{\pi^2(a|s_1^1)\sigma^2(s_1^1,a)}{b(a|s_1^1)} + \sum_a \sum_{s_j^2} \sum_{a'} \frac{\gamma^2\pi^2(a|s_1^1)P(s_j^2|s_1^1,a)\pi^2(a|s_j^2)\sigma^2(s_j^2,a')}{b(a|s_1^1)b(a'|s_j^2)}$$

$$+ \sum_s \lambda_s \left( \sum_a b(a|s) - 1 \right)$$

**Step 5 (Solving for KKT condition):** Now we want to get the KKT condition for the Lagrangian function $L(\lambda,\mathbf{b})$ as follows:

$$\nabla_{\lambda_s} L(\lambda,\mathbf{b}) = \sum_a b(a|s) - 1 \tag{20}$$

$$\nabla_{b(a|s_1^1)} L(\lambda,\mathbf{b}) = -\frac{\pi^2(a|s_1^1)\sigma^2(s_1^1,a)}{b(a|s_1^1)^2} - \gamma^2\pi^2(a|s_1^1)\sum_{s_j^2}\sum_{a'} \frac{P(s_j^2|s_1^1,a)\pi^2(a'|s_j^2)\sigma^2(s_j^2,a')}{b^2(a|s_1^1)b(a'|s_j^2)} + \lambda_{s_1^1} \tag{21}$$

$$\nabla_{b(a'|s_j^2)} L(\lambda,\mathbf{b}) = -\sum_{a'} \gamma^2\pi^2(a|s_1^1)\frac{P(s_j^2|s_1^1,a)\pi^2(a'|s_j^2)\sigma^2(s_j^2,a')}{b(a|s_1^1)b^2(a'|s_j^2)} + \lambda_{s_j^2} \tag{22}$$

Setting (22) equal to 0, we obtain:

$$\lambda_{s_j^2} = \sum_a \gamma^2\pi^2(a|s_1^1)\frac{P(s_j^2|s_1^1,a)\pi^2(a'|s_j^2)\sigma^2(s_j^2,a')}{b(a|s_1^1)b^2(a'|s_j^2)} \tag{23}$$

$$\Longrightarrow b(a|s_j^2) = \sqrt{\sum_a \gamma^2\pi^2(a|s_1^1)\frac{P(s_j^2|s_1^1,a)\pi^2(a'|s_j^2)\sigma^2(s_j^2,a')}{b(a|s_1^1)\lambda_{s_j^2}}} \tag{24}$$

Finally, we eliminate $\lambda_{s_j^2}$ by setting (20) to 0 and using the fact that $\sum_a b(a|s_j^2) = 1$:

$$b^*(a|s_j^2) = \frac{\pi(a|s_j^2)\sigma(s_j^2,a)}{\sum_{a'} \pi(a'|s_j^2)\sigma(s_j^2,a')} \tag{25}$$

which gives us the optimal proportion in level 2. Similarly, setting (21) equal to 0, we obtain:

$$\lambda_{s_1^1} = \frac{\pi^2(a|s_1^1)\sigma^2(s_1^1,a)}{b^2(a|s_1^1)} + \gamma^2\pi^2(a|s_1^1)\sum_{s_j^2}\sum_{a'}\frac{P(s_j^2|s_1^1,a)\pi^2(a'|s_j^2)\sigma^2(s_j^2,a')}{b^2(a|s_1^1)b(a'|s_j^2)}$$

$$\implies b(a|s_1^1) = \sqrt{\frac{\pi^2(a|s_1^1)\sigma^2(s_1^1,a)}{\lambda_{s_1^1}} + \gamma^2\pi^2(a|s_1^1)\sum_{s_j^2}\sum_{a'}\frac{P(s_j^2|s_1^1,a)\pi^2(a'|s_j^2)\sigma^2(s_j^2,a')}{\lambda_{s_1^1}b(a'|s_j^2)}}$$

$$b(a|s_1^1) = \frac{1}{\sqrt{\lambda_{s_1^1}}}\sqrt{\pi^2(a|s_1^1)\sigma^2(s_1^1,a) + \gamma^2\pi^2(a|s_1^1)\sum_{s_j^2}\sum_{a'}\frac{P(s_j^2|s_1^1,a)\pi^2(a'|s_j^2)\sigma^2(s_j^2,a')}{b(a'|s_j^2)}}$$

$$\implies b(a|s_1^1) \overset{(a)}{=} \sqrt{\frac{\pi^2(a|s_1^1)\sigma^2(s_1^1,a)}{\lambda_{s_1^1}} + \gamma^2\pi^2(a|s_1^1)\sum_{s_j^2}\sum_{a'}\frac{P(s_j^2|s_1^1,a)\pi^2(a'|s_j^2)\sigma^2(s_j^2,a')}{\lambda_{s_1^1}b(a'|s_j^2)}}$$

$$b^*(a|s_1^1) = \frac{1}{\sqrt{\lambda_{s_1^1}}}\sqrt{\pi^2(a|s_1^1)\sigma^2(s_1^1,a) + \gamma^2\pi^2(a|s_1^1)\sum_{s_j^2}\sum_{a'}P(s_j^2|s_1^1,a)B^2(s_j^2)}$$

where, $(a)$ follows by plugging in the definition of $b(a'|s_j^2)$ and substituting $B(s_j^2) = \sum_a \pi(a|s_j^2)\sigma(s_j^2,a)$. This concludes the proof for the optimal sampling in the 2-depth stochastic tree MDP $\mathbf{T}$. $\qquad\square$

# C   THREE STATE DETERMINISTIC TREE SAMPLING

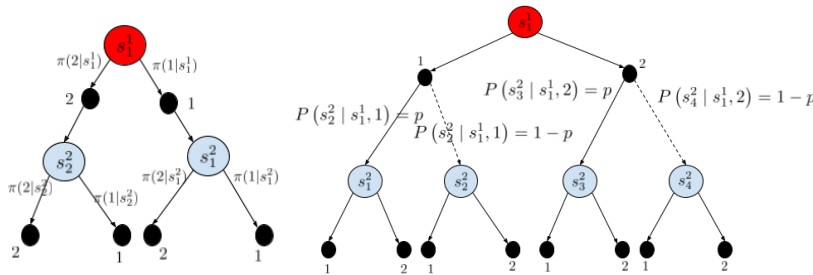

Figure 4: (Left) Deterministic 2-depth Tree. (Right) Stochastic 2-Depth Tree with varying model.

Consider the 2-depth, 2-action deterministic tree MDP $\mathbf{T}$ in Figure 4 (left) where we have equal target probabilities $\pi(1|s_1^1) = \pi(2|s_1^1) = \pi(1|s_1^2) = \pi(2|s_1^2) = \pi(1|s_2^2) = \pi(2|s_2^2) = \frac{1}{2}$. The variance is given by $\sigma^2(s_1^1,1) = 400$, $\sigma^2(s_1^1,2) = 600$, $\sigma^2(s_1^2,1) = 400$, $\sigma^2(s_1^2,2) = 400$, $\sigma^2(s_2^2,1) = 4$, $\sigma^2(s_2^2,2) = 4$. So the left sub-tree has lesser variance than right sub-tree. Let discount factor $\gamma = 1$. Then we get the optimal sampling behavior policy as follows:

$$b^*(1|s_1^2) \propto \pi(1|s_1^2)\sigma(1|s_1^2) = \frac{1}{2}\cdot 20 = 10, \qquad b^*(2|s_1^2) \propto \pi(2|s_1^2)\sigma(2|s_1^2) = \frac{1}{2}\cdot 20 = 10$$

$$b^*(1|s_2^2) \propto \pi(1|s_2^2)\sigma(1|s_2^2) = \frac{1}{2}\cdot 2 = 1, \qquad b^*(2|s_2^2) \propto \pi(2|s_2^2)\sigma(2|s_2^2) = \frac{1}{2}\cdot 2 = 1,$$

$$B(s_1^2) = \pi(1|s_1^2)\sigma(1|s_1^2) + \pi(2|s_1^2)\sigma(2|s_1^2) = 20, \qquad B(s_2^2) = \pi(1|s_2^2)\sigma(1|s_2^2) + \pi(2|s_2^2)\sigma(2|s_2^2) = 2$$

$$b^*(1|s_1^1) \propto \sqrt{\pi^2(1|s_1^1)\left[\sigma^2(s_1^1,1) + \gamma^2\sum_{s_j^2}P(s_j^2|s_1^1,1)B^2(s_j^2)\right]}$$

$$= \sqrt{\pi^2(1|s_1^1)\sigma^2(s_1^1,1) + \gamma^2\pi^2(1|s_1^1)P(s_1^2|s_1^1,1)B^2(s_1^2) + \gamma^2\pi^2(2|s_1^1)P(s_2^2|s_1^1,1)B^2(s_2^2)}$$

$$\overset{(a)}{=} \sqrt{400\cdot\frac{1}{4} + \frac{1}{4}\cdot 1\cdot 400 + \frac{1}{4}\cdot 0\cdot 4} \approx 14$$

$$b^*(2|s_1^1) \propto \sqrt{\pi^2(2|s_1^1)\left[\sigma^2(s_1^1,2) + \gamma^2 \sum_{s_j^2} P(s_j^2|s_1^1,2)B^2(s_j^2)\right]}$$

$$= \sqrt{\pi^2(2|s_1^1)\sigma^2(s_1^1,2) + \gamma^2\pi^2(2|s_1^1)P(s_1^2|s_1^1,2)B^2(s_1^2) + \gamma^2\pi^2(2|s_1^1)P(s_2^2|s_1^1,2)B^2(s_2^2)}$$

$$\overset{(b)}{=} \sqrt{600 \cdot \frac{1}{4} + \frac{1}{4} \cdot 0 \cdot 400 + \frac{1}{4} \cdot 1 \cdot 4} \approx 12$$

where, $(a)$ follows because $P(s_2^2|s_1^1,1) = 0$ and $(b)$ follows $P(s_1^2|s_1^1,2) = 0$. Note that $b(1|s_1^1)$ and $b(2|s_1^1)$ are un-normalized values. After normalization we can show that $b(1|s_1^1) > b(2|s_1^1)$. Hence the right sub-tree with higher variance will have higher proportion of pulls.

# D  THREE STATE STOCHASTIC TREE SAMPLING WITH VARYING MODEL

In this tree MDP $\mathbf{T}$ in Figure 4 (right) we have $P(s_1^2|s_1^1,1) = p$, $P(s_1^2|s_1^1,2) = 1-p$ and $P(s_2^2|s_1^1,1) = p$, $P(s_2^2|s_1^1,2) = 1-p$. Plugging this transition probabilities from the result of Lemma 1 we get

$$b^*(a|s_j^2) \propto \pi(a|s_j^2)\sigma(s_j^2,a), \quad \text{for } j \in \{1,2,3,4\}$$

$$b^*(1|s_1^1) \propto \sqrt{\pi^2(1|s_1^1)\left[\sigma^2(s_1^1,1) + \gamma^2 p B^2(s_1^2) + \gamma^2(1-p)B^2(s_2^2)\right]},$$

$$b^*(2|s_1^1) \propto \sqrt{\pi^2(2|s_1^1)\left[\sigma^2(s_1^1,2) + \gamma^2 p B^2(s_3^2) + \gamma^2(1-p)B^2(s_4^2)\right]}$$

where, $B(s_j^2) = \sum_a \pi(a|s_j^2)\sigma(s_j^2,a)$. Now if $p \gg 1-p$, then we only need to consider the variance of state $s_1^2$ when estimating the sampling proportion for states $s_1^2$ and $s_3^2$ as

$$b^*(1|s_1^1) \propto \sqrt{\pi^2(1|s_1^1)\left[\sigma^2(s_1^1,1) + \gamma^2 p B^2(s_1^2)\right]}, \qquad b^*(2|s_1^1) \propto \sqrt{\pi^2(2|s_1^1)\left[\sigma^2(s_1^1,2) + \gamma^2 p B^2(s_3^2)\right]}.$$

**Remark 1. (Transition Model Matters)** Observe that the main goal of the optimal sampling proportion in Lemma 1 is to reduce the variance of the estimate of the return. However, the sampling proportion is not geared to estimate the model $\widehat{P}$ well. An interesting extension to combine the optimization problem in Lemma 1 with some model estimation procedure as in Zanette et al. [2019], Agarwal et al. [2019], Wagenmaker et al. [2021] to derive the optimal sampling proportion.

# E  MULTI-LEVEL STOCHASTIC TREE MDP FORMULATION

**Theorem 1. (Restatement)** *Assume the underlying MDP is an $L$-depth tree MDP as defined in Definition 1. Let the estimated return of the starting state $s_1^1$ after $n$ state-action-reward samples be defined as $Y_n(s_1^1)$. Note that the $v^\pi(s_1^1)$ is the expectation of $Y_n(s_1^1)$ under Assumption 1. Let $\mathcal{D}$ be the observed data over $n$ state-action-reward samples. To minimize the MSE, $\mathbb{E}_{\mathcal{D}}[(Y_n(s_1^1)) - \mu(Y_n(s_1^1)))^2]$, the optimal sampling proportions for any arbitrary state is given by:*

$$b^*(a|s_i^\ell) \propto \sqrt{\pi^2(a|s_i^\ell)\left[\sigma^2(s_i^\ell,a) + \gamma^2 \sum_{s_j^{\ell+1}} P(s_j^{\ell+1}|s_i^\ell,a)B^2(s_j^{\ell+1})\right]},$$

*where, $B(s_j^2)$ is the normalization factor defined as follows:*

$$B(s_i^\ell) = \sum_a \sqrt{\pi^2(a|s_i^\ell)\left(\sigma^2(s_i^\ell,a) + \gamma^2 \sum_{s_j^{\ell+1}} P(s_j^{\ell+1}|s_i^\ell,a)B^2(s_j^{\ell+1})\right)}$$

*Proof.* **Step 1 (Base case for Level $L$ and $L-1$):** The proof of this theorem follows from induction. First consider the last level $L$ containing the leaf states. An arbitrary state in the last level is denoted by $s_i^L$. Then we have the estimate of the

expected return from the state $s_i^L$ as

$$Y_n(s_1^1) = \sum_{a=1}^A \pi(a|s_1^1)\left(\frac{1}{T_n(s_1^1,a)}\sum_{h=1}^{T_n(s_1^1,a)} R_h(s_1^1,a) + \gamma \sum_{s_j^{\ell+1}} P(s_j^{\ell+1}|s_1^1,a)Y_n(s_j^2)\right)$$

$$= \sum_{a=1}^A \pi(a|s_1^1)\left(\hat{\mu}(s_1^1,a) + \gamma\sum_{s_j^{\ell+1}} P(s_j^{\ell+1}|s_1^1,a)Y_n(s_j^2)\right)$$

Observe that for the leaf-state the $Y_n(s_i^L)$ the transition probability to next states $P(s_j^{L+1}|s_i^L,a) = 0$ for any action $a$. So $Y_n(s_i^L) = \sum_{a=1}^A \left(\pi(a|s_i^L)\hat{\mu}(s_i^L,a)\right)$ which matches the bandit setting. We define an estimator $Y_n(s_i^\ell)$ as defined in (2). Following the previous derivation in Lemma 1 we can show its expectation is given as:

$$\mathbb{E}[Y_n(s_i^L)] = \sum_a \frac{\pi(a|s_i^L)}{T_n(s_i^L,a)} \sum_{h=1}^{T_n(s_i^L,a)} \mathbb{E}[R_h(s_i^L,a)] = \sum_a \pi(a|s_i^L)\mu(s_i^L,a) = v^\pi(s_i^L).$$

$$\mathbf{Var}[Y_n(s_i^L)] = \sum_a \frac{\pi^2(a|s_i^L)}{T_n^2(s_i^L,a)} \sum_{h=1}^{T_n(s_i^L,a)} \mathbf{Var}[R_h(s_i^L,a)] = \sum_a \frac{\pi^2(a|s_i^L)\sigma^2(s_i^L,a)}{T_n(s_i^L,a)}$$

Now consider the second last level $L-1$ containing the leaves. An arbitrary state in the last level is denoted by $s_i^{L-1}$. Then we have the expected return from the state $s_i^{L-1}$ as follows:

$$Y_n(s_i^{L-1}) = \sum_a \pi(a|s_i^{L-1})\left(\frac{1}{T_n(s_i^{L-1},a)}\sum_{h=1}^{T_n(s_i^{L-1},a)} R_h(s_i^{L-1},a) + \gamma\sum_{s_j^L} P(s_j^L|s_i^{L-1},a)Y_n(s_j^L)\right)$$

$$= \sum_a \pi(a|s_i^{L-1})\left(\hat{\mu}(s_i^{L-1},a) + \gamma\sum_{s_j^L} P(s_j^L|s_i^{L-1},a)Y_n(s_j^L)\right).$$

Then for the estimator $Y_n(s_i^{L-1})$ we can show that its expectation is given as follows:

$$\mathbb{E}[Y_n(s_i^{L-1})] = \sum_a \pi(a|s_i^{L-1})\left[\frac{1}{T_n(s_i^{L-1},a)}\sum_{h=1}^{T_n(s_i^{L-1},a)} \mathbb{E}[R_h(s_i^{L-1},a)] + \gamma\sum_{s_j^L} P(s_j^L|s_i^{L-1},a)\mathbb{E}[Y_n(s_j^L)]\right]$$

$$= \sum_a \pi(a|s_i^{L-1})\left[\mu(s_i^{L-1},a) + \gamma\sum_{s_j^L} P(s_j^L|s_i^{L-1},a)v_n^\pi(Y(s_j^L))\right] = v_n^\pi(s_i^{L-1}).$$

$$\mathbf{Var}[Y_n(s_i^{L-1})] = \sum_a \pi^2(a|s_i^{L-1})\left[\frac{1}{T_n^2(s_i^{L-1},a)}\sum_{h=1}^{T_n(s_i^{L-1},a)} \mathbf{Var}[R_h(s_i^{L-1},a)] + \gamma^2\sum_{s_j^L} P^2(s_j^L|s_i^{L-1},a)\mathbf{Var}[Y_n(s_j^L)]\right]$$

$$\stackrel{(a)}{=} \sum_a \pi^2(a|s_i^{L-1})\left[\frac{\sigma^2(s_i^{L-1},a)}{T_n(s_i^{L-1},a)} + \gamma^2\sum_{s_j^L} P^2(s_j^L|s_i^{L-1},a)\mathbf{Var}[Y_n(s_j^L)]\right]$$

where, $(a)$ follows as $\sum_{h=1}^{T_n(s_i^{L-1},a)} \mathbf{Var}[R_h(s_i^{L-1},a)] = T_n(s_i^{L-1},a)\sigma^2(s_i^{L-1},a)$. Observe that in state $s_i^{L-1}$ we want to reduce the variance $\mathbf{Var}[Y_n(s_i^{L-1})]$. Also the optimal proportion $b^*(a|s_i^{L-1})$ to reduce variance at state $s_i^{L-1}$ cannot differ from the optimal $b^*(a|s_i^L)$ of level $L$ which reduces the variance of $b^*(a|s_i^L)$. Hence, we can follow the same optimization as done in Lemma 1 and show that the optimal sampling proportion in state $s_i^{L-1}$ is given by

$$b^*(a|s_j^L) \propto \pi(a|s_j^L)\sigma(s_j^L,a)$$

$$b^*(a|s_i^{L-1}) \stackrel{(a)}{\propto} \sqrt{\pi^2(a|s_i^{L-1})\left[\sigma^2(s_i^{L-1},a) + \gamma^2\sum_{s_j^L} P(s_j^L|s_i^{L-1},a)B_{s_j^L}^2\right]}$$

where, in $(a)$ the $s_j^L$ is the state that follows after taking action $a$ at state $s_i^{L-1}$ and $B_{s_j^L}$ is defined in (4). This concludes the base case of the induction proof. Now we will go to the induction step.

**Step 2 (Induction step for Arbitrary Level $\ell$):** We will assume that all the sampling proportion till level $\ell + 1$ from $L$ which is

$$b^*(a|s_i^{\ell+1}) \propto \sqrt{\pi^2(a|s_i^\ell)\left[\sigma^2(s_i^\ell, a) + \gamma^2 \sum_{s_j^{\ell+1}} P(s_j^{\ell+1}|s_i^\ell, a) B_{s_j^{\ell+1}}^2\right]}$$

is true. For the arbitrary level $\ell + 1$ we will use dynamic programming. We build up from the leaves (states $s_i^L$) up to estimate $b^*(a|s_i^{\ell+1})$. Then we need to show that at the previous level $\ell$ we get a similar recursive sampling proportion. We first define the estimate of the return from an arbitrary state $s_i^\ell$ in level $\ell$ after $n$ timesteps as follows:

$$Y_n(s_i^\ell) = \sum_a \pi(a|s_i^\ell) \left(\frac{1}{T_n(s_i^\ell, a)} \sum_{h=1}^{T_n(s_i^\ell, a)} R_h(s_i^\ell, a) + \gamma \sum_{s_j^{\ell+1}} P(s_j^{\ell+1}|s_i^\ell, a) Y_n(s_j^{\ell+1})\right)$$

Then we have the expectation of $Y_n(s_i^\ell)$ as follows:

$$\mathbb{E}[Y_n(s_i^\ell)] \stackrel{(a)}{=} \sum_a \pi(a|s_i^\ell) \left(\mu(s_i^\ell, a) + \gamma \sum_{s_j^{\ell+1}} P(s_j^{\ell+1}|s_i^\ell, a) v_n^\pi(Y_n(s_j^{\ell+1}))\right)$$

where, in $(a)$ the $v_n^\pi(Y(s_j^{\ell+1})) = \mathbb{E}[Y_n(s_j^{\ell+1})]$. Then we can also calculate the variance of $Y_n(s_i^\ell)$ as follows:

$$\mathbf{Var}[Y_n(s_i^\ell)] = \sum_a \pi^2(a|s_i^\ell) \left[\frac{\sigma^2(s_i^\ell, a)}{T_n(s_i^\ell, a)} + \gamma^2 \sum_{s_j^{\ell+1}} P(s_j^{\ell+1}|s_i^\ell, a) \mathbf{Var}(Y_n(s_j^{\ell+1}))\right].$$

Again observe that the goal is to minimize the variance $\mathbf{Var}[Y_n(s_i^\ell)]$. Then following the same steps in Lemma 1 we can have the optimization problem to reduce the variance which results in the following optimal sampling proportion:

$$b^*(a|s_i^\ell) \propto \sqrt{\pi^2(a|s_i^\ell)\left[\sigma^2(s_i^\ell, a) + \gamma^2 \sum_{s_j^{\ell+1}} P(s_j^{\ell+1}|s_i^\ell, a) B^2(s_j^{\ell+1})\right]}$$

where in the last equation we use $B_{s_j^{\ell+1}}$ which is defined in (4). Again we can apply Lemma 1 because the optimal proportion $b^*(a|s_i^\ell)$ to reduce variance at state $s_i^\ell$ cannot differ from the optimal $b^*(a|s_i^{\ell+1})$ of level $\ell + 1$ to $L$ which reduces the variance of $b^*(a|s_j^{\ell+1})$ to $b^*(a|s_m^L)$.

**Step 3 (Starting state $s_1^1$:)** Finally we conclude by stating that the starting state $s_1^1$ we have the estimate of the return as follows:

$$Y_n(s_1^1) = \sum_a \pi(a|s_1^1) \left(\frac{1}{T_L^K(s_1^1, a)} \sum_{h=1}^{T_n(s_1^1, a)} R_h(s_1^1, a) + \gamma \sum_{s_j^2} P(s_j^2|s_1^1, a) Y_n(s_j^2)\right).$$

Then we have the expectation of $Y_n(s_1^1)$ as follows:

$$\mathbb{E}[Y_n(s_1^1)] \stackrel{(a)}{=} \sum_a \pi(a|s_1^1) \left(\mu(s_1^1, a) + \gamma \sum_{s_j^2} P(s_j^2|s_1^1, a) v_n^\pi(Y_n(s_j^2))\right)$$

where, in $(a)$ the $v_n^\pi(s_j^1) = \mathbb{E}[Y_n(s_1^1)]$. Then we can also calculate the variance of $Y_n(s_1^1)$ as follows:

$$\mathbf{Var}[Y_n(s_1^1)] = \sum_a \pi^2(a|s_1^1) \left[\frac{\sigma^2(s_1^1, a)}{T_n(s_1^1, a)} + \gamma^2 \sum_{s_j^2} P(s_j^2|s_1^1, a) \mathbf{Var}[Y_n(s_j^2)]\right]$$

Then from the previous step 2 we can show that to reduce the variance $\mathbf{Var}[Y_n(s_1^1)]$ we should have the sampling proportion at $s_1^1$ as follows:

$$b^*(a|s_1^1) \propto \sqrt{\pi^2(a|s_1^1)\left[\sigma^2(s_1^1,a) + \gamma^2 \sum_{s_j^2} P(s_j^2|s_1^1,a)B^2(s_j^2)\right]}$$

where, in $(a)$ the $s_j^2$ is the state that follows after taking action $a$ at state $s_1^1$, and $B_{s_j^1}$ is defined in (4).   □

# F    MSE OF THE ORACLE IN TREE MDP

**Proposition 2.** *(Restatement) Let there be an oracle which knows the state-action variances and transition probabilities of the $L$-depth tree MDP $\mathbf{T}$. Let the oracle take actions in the proportions given by Theorem 1. Let $\mathcal{D}$ be the observed data over $n$ state-action-reward samples such that $n = KL$. Then the oracle suffers a MSE of*

$$\mathcal{L}_n^*(b) = \sum_{\ell=1}^{L} \left[ \frac{B^2(s_i^\ell)}{T_L^{*,K}(s_i^\ell)} + \gamma^2 \sum_a \pi^2(a|s_i^\ell) \sum_{s_j^{\ell+1}} P(s_j^{\ell+1}|s_i^\ell,a) \frac{B^2(s_j^{\ell+1})}{T_L^{*,K}(s_j^{\ell+1})} \right].$$

*where, $T_L^{*,K}(s_i^\ell)$ denotes the optimal state samples of the oracle at the end of episode $K$.*

*Proof.* **Step 1 (Arbitrary episode $k$):** First we start at an arbitrary episode $k$. For brevity we drop the index $k$ in our notation in this step. Let $n'$ be the total number of samples collected up to the $k$-th episode. We define the estimate of the return from starting state after total of $n'$ samples as

$$Y_{n'}(s_1^1) = \sum_a \pi(a|s_1^1) \left( \frac{1}{T_{n'}(s_1^1,a)} \sum_{h=1}^{T_{n'}(s_1^1,a)} R_h(s_1^1,a) + \gamma \sum_{s_j^2} P(s_j^2|s_1^1,a)Y_{n'}(s_j^2) \right).$$

Then we define the MSE as

$$\mathbb{E}_{\mathcal{D}}\left[ \left( Y_{n'}(s_1^1) - \mu(Y_{n'}(s_1^1)) \right)^2 \right] = \mathbf{Var}(Y_{n'}(s_1^1)) + \text{bias}^2(Y_{n'}(s_1^1)).$$

Again it can be shown using Theorem 1 that once all the state-action pairs are visited once we have the bias to be zero. So we want to reduce the variance $\mathbf{Var}(Y_{n'}(s_1^1))$. Note that the variance is given by

$$\mathbf{Var}[Y_{n'}(s_1^1)] = \sum_a \pi^2(a|s_1^1) \left[ \underbrace{\frac{\sigma^2(s_1^1,a)}{T_{n'}(s_1^1,a)}}_{\text{Variance of } s_1^1} + \gamma^2 \sum_{s_j^2} P(s_j^2|s_1^1,a) \underbrace{\mathbf{Var}[Y_{n'}(s_j^2)]}_{\text{Variance of } s_j^2 \text{ in level 2}} \right]. \tag{26}$$

Then we can show from the result of Theorem 1 that to minimize the $\mathbf{Var}[Y_{n'}(s_1^1)]$ the optimal sampling proportion for the level 0 is given by:

$$b^*(a|s_1^1) = \frac{\sqrt{\sum_{s_j^2} \pi^2(a|s_1^1)\left[\sigma^2(s_1^1,a) + \gamma^2 P(s_j^2|s_1^1,a)B_{s_j^2}^2\right]}}{B(s_1^1)}$$

where, $s_j^2$ are the next states of the state $s_1^1$, and $B_{s_1^1}$ as defined in (4). Let the optimal number of samples of the state-action pair $(s_i^\ell,a)$ that an oracle can take in the $k$-th episode be denoted by $T_L^{*,K}(s_i^\ell,a)$. Also let the total number of samples taken in state $s_1^1$ be $T_L^{*,K}(s_1^1)$. It follows then $n' = \sum_{s_j^\ell \in \mathcal{S}} T_{n'}^{*,k}(s_j^\ell)$. Then we have

$$T_{n'}^{*,k}(s_1^1,a) = \frac{\sqrt{\sum_{s_j^2} \pi^2(a|s_1^1)\left[\sigma^2(s_1^1,a) + \gamma^2 P(s_j^2|s_1^1,a)B_{s_j^2}^2\right]}}{B_{s_1^1}} T_{n'}^k(s_1^1).$$

where we define the normalization factor $B_{s_j^\ell}$ as in (4) and $T_{n'}^k(s_1^1)$ is the actual total number of times the state $s_1^1$ is visited. Plugging this back in (26) we get that

$$\mathbf{Var}[Y_{n'}(s_1^1)] = \sum_a \pi^2(a|s_1^1)\left[\frac{\sigma^2(s_1^1,a)}{T_{n'}^{*,k}(s_1^1,a)} + \gamma^2 \sum_{s_j^2} P(s_j^2|s_1^1,a)\mathbf{Var}[Y_{n'}(s_j^2)]\right]$$

$$= \frac{B(s_1^1)}{T_{n'}^{*,k}(s_1^1)} \sum_a \frac{\pi^2(a|s_1^1)\sigma^2(s_1^1,a)}{\sqrt{\sum_{s_j^2}\pi^2(a|s_1^1)\left[\sigma^2(s_1^1,a)+\gamma^2 P^2(s_j^2|s_1^1,a)B_{s_j^2}^2\right]}} + \gamma^2\sum_a \pi^2(a|s_1^1)\sum_{s_j^2}P(s_j^2|s_1^1,a)\mathbf{Var}[Y_{n'}(s_j^2)]$$

$$\overset{(a)}{\leq} \frac{B(s_1^1)}{T_{n'}^{*,k}(s_1^1)} \sum_a \frac{\sum_{s_j^2}\pi^2(a|s_1^1)\left[\sigma^2(s_1^1,a)+\gamma^2 P^2(s_j^2|s_1^1,a)B_{s_j^2}^2\right]}{\sqrt{\sum_{s_j^2}\pi^2(a|s_1^1)\left[\sigma^2(s_1^1,a)+\gamma^2 P(s_j^2|s_1^1,a)B_{s_j^2}^2\right]}} + \gamma^2\sum_a \pi^2(a|s_1^1)\sum_{s_j^2}P(s_j^2|s_1^1,a)\mathbf{Var}[Y_{n'}(s_j^2)]$$

$$= \frac{B_{s_1^1}}{T_{n'}^{*,k}(s_1^1)} \sum_a \sqrt{\sum_{s_j^2}\pi^2(a|s_1^1)\left[\sigma^2(s_1^1,a)+\gamma^2 P(s_j^2|s_1^1,a)B_{s_j^2}^2\right]} + \gamma^2\sum_a \pi^2(a|s_1^1)\sum_{s_j^2}P(s_j^2|s_1^1,a)\mathbf{Var}[Y_{n'}(s_j^2)]$$

$$\overset{(b)}{=} \frac{B_{s_1^1}^2}{T_{n'}^{*,k}(s_1^1)} + \gamma^2\sum_a \pi^2(a|s_1^1)\sum_{s_j^2}P(s_j^2|s_1^1,a)\underbrace{\sum_{a'}\pi^2(a'|s_j^2)\left[\frac{\sigma^2(s_j^2,a')}{T_n^k(s_j^2,a')}+\gamma^2\sum_{s_m^3}P(s_m^3|s_j^2,a')\mathbf{Var}[Y_{n'}(s_m^3)]\right]}_{\mathbf{Var}[Y_{n'}(s_j^2)]}$$

$$\overset{(c)}{\leq} \frac{B_{s_1^1}^2}{T_{n'}^{*,k}(s_1^1)} + \gamma^2\sum_a \pi^2(a|s_1^1)\sum_{s_j^2}P(s_j^2|s_1^1,a)\frac{B_{s_j^2}^2}{T_{n'}^{*,k}(s_j^2)}$$

$$\qquad + \gamma^4\sum_a \pi^2(a|s_1^1)\sum_{s_j^2}P(s_j^2|s_1^1,a)\sum_{a'}\pi^2(a'|s_j^2)\sum_{s_m^3}P(s_m^3|s_j^2,a')\mathbf{Var}[Y_{n'}(s_m^3)]$$

$$\overset{(d)}{\leq} \sum_{\ell=1}^{L}\left[\frac{B^2(s_i^\ell)}{T_{n'}^{*,k}(s_i^\ell)} + \gamma^{2\ell}\sum_a \pi^2(a|s_i^\ell)\sum_{s_j^{\ell+1}}P(s_j^{\ell+1}|s_i^\ell,a)\frac{B^2(s_j^{\ell+1})}{T_{n'}^{*,k}(s_j^{\ell+1})}\right]$$

where, $(a)$ follows as $\gamma^2 B_{s_j^1}^2 \geq 0$, $(b)$ follows by the definition of $\mathbf{Var}[Y_{s_j^2}]$ and the definition of $B(s_1^1)$ and $T_{n'}^k(s_j^2)$ is the actual number of samples observed for $s_j^2$, $(c)$ follows by substituting the value of $T_{n'}^{*,k}(s_j^2,a') = b^*(a'|s_j^2)/B(s_j^2)$, and $(d)$ follows when unrolling the equation for $L$ times.

**Step 2 (End of $K$ episodes):** Note that the above derivation holds for an arbitrary episode $k$ which consist of $L$ step horizon from root to leaf. Hence the MSE of the oracle after $K$ episodes when running behavior policy $b$ is given as

$$\mathcal{L}_n^*(b) = \sum_{\ell=1}^{L}\left[\frac{B^2(s_i^\ell)}{T_n^{*,K}(s_i^\ell)} + \gamma^{2\ell}\sum_a \pi^2(a|s_i^\ell)\sum_{s_j^{\ell+1}}P(s_j^{\ell+1}|s_i^\ell,a)\frac{B^2(s_j^{\ell+1})}{T_n^{*,K}(s_j^{\ell+1})}\right]$$

Note that $n = \sum_a \sum_{s_i^\ell \in \mathcal{S}} T_n^{*,K}(s_i^\ell,a)$ is the total samples collected after $K$ episodes of $L$ trajectories. This gives the MSE following optimal proportion in Theorem 1.

$\square$

# G   SUPPORT LEMMAS

**Lemma 2.** *(Wald's lemma for variance) [Resnick, 2019] Let $\{\mathcal{F}_t\}$ be a filtration and $R_t$ be a $\mathcal{F}_t$-adapted sequence of i.i.d. random variables with variance $\sigma^2$. Assume that $\mathcal{F}_t$ and the $\sigma$-algebra generated by $\{R_{t'} : t' \geq t+1\}$ are independent and*

*T is a stopping time w.r.t. $\mathcal{F}_t$ with a finite expected value. If $\mathbb{E}\left[R_1^2\right] < \infty$ then*

$$\mathbb{E}\left[\left(\sum_{t'=1}^{T} R_{t'} - T\mu\right)^2\right] = \mathbb{E}[T]\sigma^2$$

**Lemma 3.** *(Hoeffding's Lemma)[Massart, 2007] Let $Y$ be a real-valued random variable with expected value $\mathbb{E}[Y] = \mu$, such that $a \leq Y \leq b$ with probability one. Then, for all $\lambda \in \mathbb{R}$*

$$\mathbb{E}\left[e^{\lambda Y}\right] \leq \exp\left(\lambda\mu + \frac{\lambda^2(b-a)^2}{8}\right)$$

**Lemma 4.** *(Concentration lemma 1) Let $V_t = R_t(s,a) - \mathbb{E}[R_t(s,a)]$ and be bounded such that $V_t \in [-\eta, \eta]$. Let the total number of times the state-action $(s,a)$ is sampled be $T$. Then we can show that for an $\epsilon > 0$*

$$\mathbb{P}\left(\left|\frac{1}{T}\sum_{t=1}^{T} R_t(s,a) - \mathbb{E}[R_t(s,a)]\right| \geq \epsilon\right) \leq 2\exp\left(-\frac{2\epsilon^2 T}{\eta^2}\right).$$

*Proof.* Let $V_t = R_t(s,a) - \mathbb{E}[R_t(s,a)]$. Note that $\mathbb{E}[V_t] = 0$. Hence, for the bounded random variable $V_t \in [-\eta, \eta]$ (by Assumption 2) we can show from Hoeffding's lemma in Lemma 3 that

$$\mathbb{E}[\exp(\lambda V_t)] \leq \exp\left(\frac{\lambda^2}{8}(\eta - (-\eta))^2\right) \leq \exp\left(2\lambda^4\eta^2\right)$$

Let $s_{t-1}$ denote the last time the state $s$ is visited and action $a$ is sampled. Observe that the reward $R_t(s,a)$ is conditionally independent. For this proof we will only use the boundedness property of $R_t(s,a)$ guaranteed by Assumption 2. Next we can bound the probability of deviation as follows:

$$
\begin{aligned}
\mathbb{P}\left(\sum_{t=1}^{T}(R_t(s,a) - \mathbb{E}[R_t(s,a)]) \geq \epsilon\right) &= \mathbb{P}\left(\sum_{t=1}^{T} V_t \geq \epsilon\right) \\
&\overset{(a)}{=} \mathbb{P}\left(e^{\lambda \sum_{t=1}^{T} V_t} \geq e^{\lambda\epsilon}\right) \\
&\overset{(b)}{\leq} e^{-\lambda\epsilon}\mathbb{E}\left[e^{-\lambda\sum_{t=1}^{T} V_t}\right] \\
&= e^{-\lambda\epsilon}\mathbb{E}\left[\mathbb{E}\left[e^{-\lambda\sum_{t=1}^{T} V_t}\big|s_{T-1}\right]\right] \\
&\overset{(c)}{=} e^{-\lambda\epsilon}\mathbb{E}\left[\mathbb{E}\left[e^{-\lambda V_T}|S_{T-1}\right]\mathbb{E}\left[e^{-\lambda\sum_{t=1}^{T-1} V_t}\big|s_{T-1}\right]\right] \\
&\leq e^{-\lambda\epsilon}\mathbb{E}\left[\exp\left(2\lambda^4\eta^2\right)\mathbb{E}\left[e^{-\lambda\sum_{t=1}^{T-1} V_t}\big|s_{T-1}\right]\right] \\
&= e^{-\lambda\epsilon}e^{2\lambda^2\eta^2}\mathbb{E}\left[e^{-\lambda\sum_{t=1}^{T-1} V_t}\right] \\
&\quad\vdots \\
&\overset{(d)}{\leq} e^{-\lambda\epsilon}e^{2\lambda^2 T\eta^2} \\
&\overset{(e)}{\leq} \exp\left(-\frac{2\epsilon^2}{T\eta^2}\right)
\end{aligned}
$$
(27)

where $(a)$ follows by introducing $\lambda \in \mathbb{R}$ and exponentiating both sides, $(b)$ follows by Markov's inequality, $(c)$ follows as $V_t$ is conditionally independent given $s_{T-1}$, $(d)$ follows by unpacking the term for $T$ times and $(e)$ follows by taking $\lambda = \epsilon/4T\eta^2$. Hence, it follows that

$$\mathbb{P}\left(\left|\frac{1}{T}\sum_{t=1}^{T} R_t(s,a) - \mathbb{E}[R_t(s,a)]\right| \geq \epsilon\right) = \mathbb{P}\left(\sum_{t=1}^{T}(R_t(s,a) - \mathbb{E}[R_t(s,a)]) \geq T\epsilon\right) \overset{(a)}{\leq} 2\exp\left(-\frac{2\epsilon^2 T}{\eta^2}\right).$$

where, $(a)$ follows by (27) by replacing $\epsilon$ with $\epsilon T$, and accounting for deviations in either direction. $\qquad\square$

**Lemma 5.** *(Concentration lemma 2)* *Let $\mu^2(s,a) = \mathbb{E}\left[R_t^2(s,a)\right]$. Let $R_t(s,a) \leq 2\eta$ and $R_t^2(s,a) \leq 4\eta^2$ for any time $t$ and following Assumption 2. Let $n = KL$ be the total budget of state-action samples. Define the event*

$$
\xi_\delta = \left( \bigcap_{s\in\mathcal{S}} \bigcap_{1\leq a\leq A, T_n(s,a)\geq 1} \left\{ \left| \frac{1}{T_n(s,a)} \sum_{t=1}^{T_n(s,a)} R_t^2(s,a) - \mu^2(s,a) \right| \leq (2\eta + 4\eta^2)\sqrt{\frac{\log(SAn(n+1)/\delta)}{2T_n(s,a)}} \right\} \right) \cap
$$

$$
\left( \bigcap_{s\in\mathcal{S}} \bigcap_{1\leq a\leq A, T_n(s,a)\geq 1} \left\{ \left| \frac{1}{T_n(s,a)} \sum_{t=1}^{T_n(s,a)} R_t(s,a) - \mu(s,a) \right| \leq (2\eta + 4\eta^2)\sqrt{\frac{\log(SAn(n+1)/\delta)}{2T_n(s,a)}} \right\} \right) \quad (28)
$$

*Then we can show that $\mathbb{P}\left(\xi_\delta\right) \geq 1 - 2\delta$.*

*Proof.* First note that the total budget $n = KL$. Observe that the random variable $R_t^k(s,a)$ and $R_t^{(2),k}(s,a)$ are conditionally independent given the previous state $S_{t-1}^k$. Also observe that for any $\eta > 0$ we have that $R_t^k(s,a), R_t^{(2),k}(s,a) \leq 2\eta + 4\eta^2$, where $R_t^{(2),k}(s,a) = (R_t^k(s,a))^2$. Hence we can show that

$$
\mathbb{P}\left( \bigcap_{s\in\mathcal{S}} \bigcap_{1\leq a\leq A, T_n(s,a)\geq 1} \left\{ \left| \frac{1}{T_n(s,a)} \sum_{t=1}^{T_n(s,a)} R_t^2(s,a) - \mu^2(s,a) \right| \geq (2\eta + 4\eta^2)\sqrt{\frac{\log(SAn(n+1)/\delta)}{2T_n(s,a)}} \right\} \right)
$$

$$
\leq \mathbb{P}\left( \bigcup_{s\in\mathcal{S}} \bigcup_{1\leq a\leq A, T_n(s,a)\geq 1} \left\{ \left| \frac{1}{T_n(s,a)} \sum_{t=1}^{T_n(s,a)} R_t^2(s,a) - \mu^2(s,a) \right| \geq (2\eta + 4\eta^2)\sqrt{\frac{\log(SAn(n+1)/\delta)}{2T_n(s,a)}} \right\} \right)
$$

$$
\overset{(a)}{\leq} \sum_{s=1}^{S}\sum_{a=1}^{A}\sum_{t=1}^{n}\sum_{T_n(s,a)=1}^{t} 2\exp\left( -\frac{2T_n}{4(\eta^2 + \eta)^2} \cdot \frac{4(\eta^2 + \eta)^2 \log(SAn(n+1)/\delta)}{2T_n(s,a)} \right) = \delta.
$$

where, $(a)$ follows from Lemma 4. Note that in $(a)$ we have to take a double union bound summing up over all possible pulls $T_n$ from $1$ to $n$ as $T_n$ is a random variable. Similarly we can show that

$$
\mathbb{P}\left( \bigcap_{s\in\mathcal{S}} \bigcap_{1\leq a\leq A, T_n(s,a)\geq 1} \left\{ \left| \frac{1}{T_n(s,a)} \sum_{t=1}^{T_n(s,a)} R_t(s,a) - \mu(s,a) \right| \geq (2\eta + 4\eta^2)\sqrt{\frac{\log(SAn(n+1)/\delta)}{2T_n}} \right\} \right)
$$

$$
\overset{(a)}{\leq} \sum_{s=1}^{S}\sum_{a=1}^{A}\sum_{t=1}^{n}\sum_{T_n(s,a)=1}^{t} 2\exp\left( -\frac{2T_n}{4(\eta^2 + \eta)^2} \cdot \frac{4(\eta^2 + \eta)^2 \log(SAn(n+1)/\delta)}{2T_n(s,a)} \right) = \delta.
$$

where, $(a)$ follows from Lemma 4. Hence, combining the two events above we have the following bound

$$
\mathbb{P}\left(\xi_\delta\right) \geq 1 - 2\delta.
$$

$\square$

**Corollary 1.** *Under the event $\xi_\delta$ in (28) we have for any state-action pair in an episode $k$ the following relation with probability greater than $1 - \delta$*

$$
|\widehat{\sigma}_t^k(s,a) - \sigma(s,a)| \leq (2\eta + 4\eta^2)\sqrt{\frac{\log(SAn(n+1)/\delta)}{2T_L^K(s,a)}}.
$$

*where, $T_L^K(s,a)$ is the total number of samples of the state-action pair $(s,a)$ till episode $k$.*

*Proof.* Observe that the event $\xi_\delta$ bounds the sum of rewards $R_t^k(s,a)$ and squared rewards $R_t^{k,(2)}(s,a)$ for any $T_L^K(s,a) \geq 1$. Hence we can directly apply the Lemma 5 to get the bound. $\square$

**Lemma 6.** (***Bound samples in level*** 2) *Suppose that, at an episode $k$, the action $p$ in state $s_i^2$ in a 2-depth* **T** *is under-pulled relative to its optimal proportion. Then we can lower bound the actual samples $T_L^K(s_i^2, p)$ with respect to the optimal samples $T_L^{*,K}(s_i^2, p)$ with probability $1 - \delta$ as follows*

$$T_L^K(s_i^2, p) \geq T_L^{*,K}(s_i^2, p) - 4cb^*(p|s_i^2)\frac{\sqrt{\log(H/\delta)}}{B(s_i^2)b_{\min}^{*,3/2}(s_i^2)}\sqrt{T_L^K(s_i^2)} - 4Ab^*(p|s_i^2),$$

*where $B(s_i^2)$ is defined in* (4)*, $c = (\eta + \eta^2)/\sqrt{2}$, and $H = SAn(n+1)$.*

*Proof.* **Step 1 (Properties of the algorithm):** Let us first define the confidence interval term for $(s, a)$ at time $t$ as

$$U_t^k(s, a) = 2c\sqrt{\frac{\log(H/\delta)}{T_t^k(s_i^2, a)}} \tag{29}$$

where, $c = (\eta + \eta^2)/\sqrt{2}$, and $H = SAn(n+1)$. Also note that on $\xi_\delta$ using Corollary 1 we have

$$\widehat{\sigma}_t^k(s_i^2, a) \overset{(a)}{\leq} \sigma(s_i^2, a) + U_t^k(s, a) \implies \widehat{\sigma}_t^{(2),k}(s_i^2, a) \leq \sigma^2(s_i^2, a) + 2\sigma(s_i^2, a)U_t^k(s, a) + U_t^{(2),k}(s, a)$$

$$= \sigma^2(s_i^2, a) + 4\sigma c\sqrt{\frac{\log(H/\delta)}{T_t^k(s_i^2, a)}} + 4c^2\frac{\log(H/\delta)}{T_t^k(s_i^2, a)}$$

$$\overset{(b)}{\leq} \sigma^2(s_i^2, a) + 4dc^2\sqrt{\frac{\log(H/\delta)}{T_t^k(s_i^2, a)}} \tag{30}$$

where, $(a)$ follows from Corollary 1, and $(b)$ follows for some constant $d > 0$ and noting that $\sqrt{\frac{\log(H/\delta)}{T_t^k(s_i^2, a)}} > \frac{\log(H/\delta)}{T_t^k(s_i^2, a)}$ and $c^2 > c$. Let $a$ be an arbitrary action in state $s_i^2$. Recall the definition of the upper bound used in ReVar when $t > 2SA$:

$$\overline{U}_{t+1}^k(a|s_i^2) = \frac{\widehat{b}_t^k(a|s_i^2)}{T_t^k(s_i^2, a)} = \frac{\sqrt{\pi^2(a|s_i^2)\widehat{\sigma u}_t^{(2),k}(s_i^2, a)}}{T_t^k(s_i^2, a)} = \frac{\sqrt{\pi^2(a|s_i^2)\left(\widehat{\sigma}_t^{(2),k}(s_i^2, a) + 4dc^2\sqrt{\frac{\log(H/\delta)}{T_t^k(s_i^2, a)}}\right)}}{T_t^k(s_i^2, a)}$$

Under the good event $\xi_\delta$ using Corollary 1, we obtain the following upper and lower bounds for $\overline{U}_{t+1}^k(a|s_i^2)$:

$$\frac{\sqrt{\pi^2(a|s_i^2)\sigma^2(s_i^2, a)}}{T_t^k(s_i^2, a)} \overset{(a)}{\leq} \overline{U}_{t+1}^k(a|s_i^2) \overset{(b)}{\leq} \frac{\sqrt{\pi^2(a|s_i^2)\left(\sigma^2(s_i^2, a) + 8dc^2\sqrt{\frac{\log(H/\delta)}{T_t^k(s_i^2, a)}}\right)}}{T_t^k(s_i^2, a)} \tag{31}$$

where, $(a)$ follows as $\sigma^2(s_i^2, a) \leq \widehat{\sigma}_t^{(2),k}(s_i^2, a) + 4dc^2\sqrt{\log(H/\delta)/T_k^t(s_i^2, a)}$ and $(b)$ follows as $\widehat{\sigma}_t^{(2),k}(s_i^2, a) + 4dc^2\sqrt{\log(H/\delta)/T_k^t(s_i^2, a)} \leq \widehat{\sigma}_t^{(2),k}(s_i^2, a) + 8dc^2\sqrt{\log(H/\delta)/T_k^t(s_i^2, a)}$. Let ReVar chooses to pull action $m$ at $t + 1 > 2SA$ in $s_i^L$ for the last time. Then we have that for any action $p \neq m$ the following:

$$\overline{U}_{t+1}^k(p|s_i^2) \leq \overline{U}_{t+1}^k(m|s_i^2).$$

Recall that $T_t^k(s_i^2, m)$ is the last time the action $m$ is sampled. Hence, $T_t^k(s_i^2, m) = T_L^K(s_i^2, m) - 1$ because we are sampling action $m$ again in time $t + 1$. Note that $T_L^K(s_i^2, m)$ is the total pulls of action $m$ at the end of time $n$. It follows from (31) then

$$\overline{U}_{t+1}^k(m|s_i^2) \leq \frac{\sqrt{\pi^2(m|s_i^2)\left(\sigma^2(s_i^2, m) + 8dc^2\sqrt{\frac{\log(H/\delta)}{T_t^k(s_i^2, m)}}\right)}}{T_t^k(s_i^2, m)} = \frac{\sqrt{\pi^2(m|s_i^2)\left(\sigma^2(s_i^2, m) + 8dc^2\sqrt{\frac{\log(H/\delta)}{T_t^k(s_i^2, m)-1}}\right)}}{T_t^k(s_i^2, m) - 1}.$$

Let $p$ be the arm in state $s_i^2$ that is under-pulled. Recall that $T_L^K(s_i^2) = \sum_a T_L^K(s_i^2, a)$. Using the lower bound in (31) and the fact that $T_t^k(s_i^2, p) \leq T_L^K(s_i^2, p)$, we may lower bound $I_{t+1}^k(p|s_i^2)$ as

$$\overline{U}_{t+1}^k(p|s_i^2) \geq \frac{\sqrt{\pi^2(p|s_i^2)\sigma^2(s_i^2, p)}}{T_t^k(s_i^2, p)} \geq \frac{\sqrt{\pi^2(p|s_i^2)\sigma^2(s_i^2, p)}}{T_L^K(s_i^2, p)}.$$

Combining all of the above we can show

$$\frac{\sqrt{\pi^2(p|s_i^2)\sigma^2(s_i^2,p)}}{T_L^K(s_i^2,p)} \leq \frac{\sqrt{\pi^2(m|s_i^2)\left(\sigma^2(s_i^2,m)+8dc^2\sqrt{\frac{\log(H/\delta)}{T_L^K(s_i^2,m)-1}}\right)}}{T_L^K(s_i^2,m)-1}. \tag{32}$$

Observe that there is no dependency on $t$, and thus, the probability that (32) holds for any $p$ and for any $m$ is at least $1-\delta$ (probability of event $\xi_\delta$).

**Step 2 (Lower bound on $T_L^K(s_i^2,p)$):** If an action $p$ is under-pulled compared to its optimal allocation without taking into account the initialization phase,i.e., $T_L^K(s_i^2,p)-2 < b(p|s_i^2)(T_n(s_i^2)-2A)$, then from the constraint $\sum_a \left(T_L^K(s_i^2,a)-2\right) = T_L^K(s_i^2)-2A$ and the definition of the optimal allocation, we deduce that there exists at least another action $m$ that is over-pulled compared to its optimal allocation without taking into account the initialization phase, i.e., $T_n^k(s_i^2,m)-2 > b(m|s_i^2)(T_L^K(s_i^2)-2SA)$.

$$
\begin{aligned}
\frac{\sqrt{\pi^2(p|s_i^2)\sigma^2(s_i^2,p)}}{T_L^K(s_i^2,p)} &\leq \frac{\sqrt{\pi^2(m|s_i^2)\left(\sigma^2(s_i^2,m)+8dc^2\sqrt{\frac{\log(H/\delta)}{T_L^K(s_i^2,m)-1}}\right)}}{T_L^K(s_i^2,m)-1} \overset{(a)}{\leq} \frac{\sqrt{\pi^2(m|s_i^2)\left(\sigma^2(s_i^2,m)+8dc^2\sqrt{\frac{\log(H/\delta)}{T_L^K(s_i^2,m)-2}}\right)}}{T_L^K(s_i^2,m)-1} \\
&\overset{(b)}{\leq} \frac{\sqrt{\pi^2(m|s_i^2)\sigma^2(s_i^2,m)}+4d\pi(m|s_i^2)c\sqrt{\frac{\log(H/\delta)}{T_L^K(s_i^2,m)-2}}}{T_L^{*,K}(s_i^2,m)} \\
&\overset{(c)}{\leq} \frac{\sqrt{\pi^2(m|s_i^2)\sigma^2(s_i^2,m)}+\left(4dc\sqrt{\frac{\log(H/\delta)}{b^*(m|s_i^2)(T_L^K(s_i^2)-2SA)+1}}\right)}{T_L^{*,K}(s_i^2,m)} \\
&\overset{(d)}{\leq} \frac{B(s_i^2)}{T_L^K(s_i^2)}+4dc\frac{\sqrt{\log(H/\delta)}}{T_L^{(3/2),K}(s_i^2)b^*(m|s_i^2)^{3/2}}+\frac{4AB(s_i^2)}{T_L^{(2),K}(s_i^2)} \\
&\overset{(e)}{\leq} \frac{B(s_i^2)}{T_L^K(s_i^2)}+4dc\frac{\sqrt{\log(H/\delta)}}{T_L^{(3/2),K}(s_i^2)b_{\min}^{*,3/2}(s_i^2)}+\frac{4AB(s_i^2)}{T_L^{(2),K}(s_i^2)}. \tag{33}
\end{aligned}
$$

where, $(a)$ follows as $T_L^K(s_i^2,m)-2 \leq T_L^K(s_i^2,m)-1$, $(b)$ follows as $T_n^{*,(k)}(s_i^2,m) \geq T_L^K(s_i^2,m)-1$ as action $m$ is over-pulled and $\sqrt{a+b} \leq \sqrt{a}+\sqrt{b}$ for $a,b>0$, $(c)$ follows as $T_L^K(s_i^2) = \sum_a T_L^K(s_i^2,a)$ and $T_n^k(s_i^2,m)-2 > b^*(m|s_i^2)(T_L^K(s_i^2)-2SA)$, $(d)$ follows by setting the optimal samples $T_L^{*,K}(s_i^2,m) = \frac{\sqrt{\pi^2(m|s_i^2)\sigma^2(s_i^2,m)}}{B(s_i^2)}T_L^K(s_i^2)$, and $(e)$ follows as $b^*(m|s_i^2) \geq b_{\min}(s_i^2)$. By rearranging (33), we obtain the lower bound on $T_L^K(s_i^2,p)$:

$$
\begin{aligned}
T_L^K(s_i^2,p) &\geq \frac{\sqrt{\pi^2(p|s_i^2)\sigma^2(s_i^2,p)}}{\frac{B(s_i^2)}{T_L^K(s_i^2)}+4dc\frac{\sqrt{\log(H/\delta)}}{T_L^{(3/2),K}(s_i^2)b_{\min}^{*,3/2}(s_i^2)}+\frac{4AB(s_i^2)}{T_L^{(2),K}(s_i^2)}} = \frac{\sqrt{\pi^2(p|s_i^2)\sigma^2(s_i^2,p)}}{\frac{B(s_i^2)}{T_L^K(s_i^2)}}\left[\frac{1}{1+4dc\frac{\sqrt{\log(H/\delta)}}{B(s_i^2)T_L^{(1/2),K}(s_i^2)b_{\min}^{*,3/2}(s_i^2)}+\frac{4A}{T_n^k(s_i^2)}}\right] \\
&\overset{(a)}{\geq} \frac{\sqrt{\pi^2(p|s_i^2)\sigma^2(s_i^2,p)}}{\frac{B(s_i^2)}{T_L^K(s_i^2)}}\left[1-4dc\frac{\sqrt{\log(H/\delta)}}{B(s_i^2)T_L^{(1/2),K}(s_i^2)b_{\min}^{*,3/2}(s_i^2)}-\frac{4A}{T_n^k(s_i^2)}\right] \\
&\geq T_L^{*,K}(s_i^2,p)-4dcb^*(p|s_i^2)\frac{\sqrt{\log(H/\delta)}}{B(s_i^2)b_{\min}^{*,3/2}(s_i^2)}\sqrt{T_L^K(s_i^2)}-4Ab^*(p|s_i^2),
\end{aligned}
$$

where in $(a)$ we use $1/(1+x) \geq 1-x$ (for $x>-1$).  □

**Lemma 7.** *(Bound samples in level 1)* *Suppose that, at an episode $k$, the action $p$ in state $s_1^1$ in a 2-depth* **T** *is under-pulled relative to its optimal proportion. Then we can lower bound the actual samples $T_L^K(s_1^1,p)$ with respect to the optimal samples $T_L^{*,K}(s_1^1,p)$ with probability $1-\delta$ as follows*

$$
\begin{aligned}
T_L^K(s_1^1,p) \geq{}& T_L^{*,K}(s_1^1,p)-4cb^*(p|s_1^1)\frac{\sqrt{\log(H/\delta)}}{B(s_1^1)b_{\min}^{*,3/2}(s_1^1)}\sqrt{T_L^K(s_1^1)}-4Ab^*(p|s_1^1) \\
&-\gamma\pi(m|s_1^1)\frac{T_L^K(s_1^1)}{B^2(s_1^1)}\sum_{s_j^2}P(s_j^2|s_1^1,m)\frac{B(s_j^2)}{b^*(m|s_j^2)}\sum_{a'}\left[T_L^{*,K}(s_j^2,a')+4cb^*(a'|s_j^2)\frac{\sqrt{\log(H/\delta)}}{b_{\min}^{*,3/2}(s_j^2)}\sqrt{T_L^K(s_1^1)}+4Ab(a'|s_j^2)\right]
\end{aligned}
$$

where $B(s_t^2)$ is defined in (4), $c = (\eta + \eta^2)/\sqrt{2}$, and $H = SAn(n+1)$.

*Proof.* **Step 1 (Properties of the algorithm):** Again note that on $\xi_\delta$ using Corollary 1 we have

$$\widehat{\sigma}_t^k(s_1^1, a) \leq \sigma(s_1^1, a) + U_t^k(s, a) \implies \widehat{\sigma}_t^{(2),k}(s_1^1, a) \leq \sigma^2(s_1^1, a) + U_t^{(2),k}(s, a) \overset{(a)}{=} \sigma^2(s_1^1, a) + 4dc^2\sqrt{\frac{\log(H/\delta)}{T_t^k(s_1^1, a)}}$$

for any action $a$ in $s_1^1$, where $(a)$ follows by the definition of $U_t^{(2),k}$ (29), some constant $d > 0$ and the same derivation as in (30). Let $a$ be an arbitrary action in state $s_1^1$. Recall the definition of the upper bound used in ReVar when $t > 2SA$:

$$\overline{U}_{t+1}^k(a|s_1^1) = \frac{\widehat{b}_t^k(a|s_1^1)}{T_t^k(s_1^1, a)} = \frac{\sqrt{\sum_{s_j^2} \pi^2(a|s_1^1)\left[\widehat{\sigma u}_t^{(2),k}(s_1^1, a) + \gamma^2 P(s_j^2|s_1^1, a)\widehat{B}_t^{(2),k}(s_j^2)\right]}}{T_t^k(s_1^1, a)}$$

$$= \frac{\sqrt{\sum_{s_j^2} \pi^2(a|s_1^1)\left[\widehat{\sigma}_t^{(2),k}(s_1^1, a) + 4dc^2\sqrt{\frac{\log(H/\delta)}{T_t^k(s_1^1,a)}} + \gamma^2 P(s_j^2|s_1^1, a)\sum_{a'}\sqrt{\pi^2(a'|s_j^2)\left(\widehat{\sigma}_t^{(2),k}(s_j^2, a') + 4dc^2\sqrt{\frac{\log(H/\delta)}{T_t^k(s_j^2,a')}}\right)}\right]}}{T_t^k(s_1^1, a)}$$

Under the good event $\xi_\delta$ using the Corollary 1, we obtain the following upper and lower bounds for $\overline{U}_{t+1}^k(a|s_1^1)$:

$$\overline{U}_{t+1}^k(a|s_1^1) \leq \frac{\sqrt{\sum_{s_j^2} \pi^2(a|s_1^1)\left[\sigma^2(s_1^1, a) + 8dc^2\sqrt{\frac{\log(H/\delta)}{T_t^k(s_1^1,a)}} + \gamma^2 P(s_j^2|s_1^1, a)\sum_{a'}\sqrt{\pi^2(a'|s_j^2)\left(\sigma^2(s_j^2, a') + 8dc^2\sqrt{\frac{\log(H/\delta)}{T_t^k(s_j^2,a')}}\right)}\right]}}{T_t^k(s_1^1, a)}$$

$$\overline{U}_{t+1}^k(a|s_1^1) \geq \frac{\sqrt{\pi^2(a|s_1^1)\sigma^2(s_1^1, a)}}{T_t^k(s_1^1, a)} \tag{34}$$

where, $(a)$ follows as $\sigma^2(s_1^1, a) \leq \widehat{\sigma}_t^{(2),k}(s_1^1, a) + 4dc^2\sqrt{\log(H/\delta)/T_k^t(s_1^1, a)}$ and $(b)$ follows as $\widehat{\sigma}_t^{(2),k}(s_1^1, a) + 4dc^2\sqrt{\log(H/\delta)/T_k^t(s_1^1, a)} \leq \widehat{\sigma}_t^{(2),k}(s_1^1, a) + 8dc^2\sqrt{\log(H/\delta)/T_k^t(s_1^1, a)}$. Let ReVar chooses to take action $m$ at $t+1$ in $s_1^1$ for the last time. Then we have that for any action $p \neq m$ the following:

$$\overline{U}_{t+1}^k(p|s_1^1) \leq \overline{U}_{t+1}^k(m|s_1^1).$$

Recall that $T_t^k(s_1^1, m)$ is the last time the action $m$ is sampled. Hence, $T_t^k(s_1^1, m) = T_L^K(s_1^1, m) - 1$ because we are sampling action $m$ again in time $t+1$. Note that $T_L^K(s_1^1, m)$ is the total pulls of action $m$ at the end of time $n$. It follows from (34)

$$\overline{U}_{t+1}^k(m|s_1^1) \leq \frac{\sqrt{\sum_{s_j^2} \pi^2(a|s_1^1)\left[\sigma^2(s_1^1, a) + 8dc^2\sqrt{\frac{\log(H/\delta)}{T_t^k(s_1^1,a)}} + \gamma^2 P(s_j^2|s_1^1, a)\sum_{a'}\sqrt{\pi^2(a'|s_j^2)\left(\sigma^2(s_j^2, a') + 8dc^2\sqrt{\frac{\log(H/\delta)}{T_t^k(s_j^2,a')}}\right)}\right]}}{T_t^k(s_1^1, a)}$$

$$\overset{(a)}{\leq} \frac{\sqrt{\sum_{s_j^2}\left(\pi^2(a|s_1^1)\sigma^2(s_1^1, a) + 8dc^2\sqrt{\frac{\log(H/\delta)}{T_t^k(s_1^1,a)}}\right)} + \gamma\pi(a|s_1^1)\sum_{s_j^2} P(s_j^2|s_1^1, a)\left[\sum_{a'}\sqrt{\pi^2(a'|s_j^2)\left(\sigma^2(s_j^2, a') + 8dc^2\sqrt{\frac{\log(H/\delta)}{T_t^k(s_j^2,a')}}\right)}\right]}{T_t^k(s_1^1, a)}$$

$$\overset{(b)}{\leq} \frac{\sqrt{\sum_{s_j^2} \pi^2(m|s_1^1)\left(\sigma^2(s_1^1, m) + 8dc^2\sqrt{\frac{\log(H/\delta)}{T_L^K(s_1^1,m)-1}}\right)}}{T_L^K(s_1^1, m) - 1}$$

$$+ \gamma\pi(a|s_1^1)\sum_{s_j^2} P(s_j^2|s_1^1, a)\sum_{a'}\left[\frac{\sqrt{\pi^2(a'|s_j^2)\left(\sigma^2(s_j^2, a') + 8dc^2\sqrt{\frac{\log(H/\delta)}{T_L^K(s_j^2,a')}}\right)}}{T_L^K(s_j^2, a') - 1}\right].$$

where, $(a)$ follows as $\sqrt{a+b} \leq \sqrt{a} + \sqrt{b}$ for $a, b > 0$ and $(b)$ follows as $T_t^k(s_1^1, a) \geq T_t^k(s_j^2, a')$ where $s_j^2$ is the next state of $s_1^1$ following action $a$.

Let $p$ be the arm in state $s_1^1$ that is under-pulled. Recall that $T_L^K(s_1^1) = \sum_a T_L^K(s_1^1, a)$. Using the lower bound in (34) and the fact that $T_t^k(s_1^1, p) \leq T_L^K(s_1^1, p)$, we may lower bound $\overline{U}_{t+1}^k(p|s_1^1)$ as

$$\overline{U}_{t+1}^k(p|s_1^1) \geq \frac{\sqrt{\pi^2(p|s_i^2)\sigma^2(s_i^2, p)}}{T_t(s_1^1, p)} \geq \frac{\sqrt{\pi^2(p|s_i^2)\sigma^2(s_i^2, p)}}{T_L^K(s_1^1, p)}.$$

Combining all of the above we can show

$$\frac{\sqrt{\pi^2(p|s_i^2)\sigma^2(s_i^2, p)}}{T_L^K(s_1^1, p)} \leq \frac{\sqrt{\sum_{s_j^2} \pi^2(m|s_1^1)\left(\sigma^2(s_1^1, m) + 8dc^2\sqrt{\frac{\log(H/\delta)}{T_L^K(s_1^1, m) - 1}}\right)}}{T_L^K(s_1^1, m) - 1}$$

$$+ \gamma\pi(m|s_1^1)\sum_{s_j^2} P(s_j^2|s_1^1, m)\sum_{a'}\left[\frac{\sqrt{\pi^2(a'|s_j^2)\left(\sigma^2(s_j^2, a') + 8dc^2\sqrt{\frac{\log(H/\delta)}{T_L^K(s_j^2, a') - 1}}\right)}}{T_L^K(s_j^2, a') - 1}\right]. \quad (35)$$

Observe that there is no dependency on $t$, and thus, the probability that (35) holds for any $p$ and for any $m$ is at least $1 - \delta$ (probability of event $\xi_\delta$).

**Step 2 (Lower bound on $T_L^K(s_1^1, p)$):** If an action $p$ is under-pulled compared to its optimal allocation without taking into account the initialization phase, i.e., $T_L^K(s_1^1, p) - 2 < b^*(p|s_1^1)(T_L^K(s_1^1) - 2A)$, then from the constraint $\sum_a \left(T_L^K(s_1^1, a) - 2\right) = T_L^K(s_1^1) - 2A$ and the definition of the optimal allocation, we deduce that there exists at least another action $m$ that is over-pulled compared to its optimal allocation without taking into account the initialization phase, i.e., $T_n^k(s_1^1, m) - 2 > b^*(m|s_1^1)(T_L^K(s_1^1) - 2SA)$.

$$\frac{\pi(p|s_1^1)\sigma(s_1^1, p)}{T_L^K(s_1^1, p)} \leq \frac{\sqrt{\sum_{s_j^2} \pi^2(m|s_1^1)\left(\sigma^2(s_1^1, m) + 8dc^2\sqrt{\frac{\log(H/\delta)}{T_L^K(s_1^1, m) - 2}}\right)}}{T_L^K(s_1^1, m) - 1}$$

$$+ \gamma\pi(m|s_1^1)\sum_{s_j^2} P(s_j^2|s_1^1, m)\sum_{a'}\left[\frac{\sqrt{\pi^2(a'|s_j^2)\left(\sigma^2(s_j^2, a') + 8dc^2\sqrt{\frac{\log(H/\delta)}{T_L^K(s_j^2, a') - 2}}\right)}}{T_L^K(s_j^2, a') - 1}\right]$$

$$\overset{(a)}{\leq} \sum_{s_j^2} \frac{\sqrt{\pi^2(m|s_1^1)\sigma^2(s_1^1, m)} + 4dc\sqrt{\frac{\log(H/\delta)}{T_L^K(s_1^1, m) - 2}}}{T_L^{*,K}(s_1^1, m)}$$

$$+ \gamma\pi(m|s_1^1)\sum_{s_j^2} P(s_j^2|s_1^1, m)\sum_{a'}\left[\frac{\sqrt{\pi^2(a'|s_j^2)\sigma^2(s_j^2, a')} + 4dc\sqrt{\frac{\log(H/\delta)}{T_L^K(s_j^2, a') - 2}}}{T_L^{*,K}(s_j^2, a')}\right]$$

$$\overset{(b)}{\leq} \sum_{s_j^2} \frac{\sqrt{\pi^2(m|s_1^1)\sigma^2(s_1^1, m)} + \left(4dc\sqrt{\frac{\log(H/\delta)}{b^*(m|s_1^1)(T_L^K(s_1^1) - 2SA) + 1}}\right)}{T_L^{*,K}(s_1^1, m)}$$

$$+ \gamma\pi(m|s_1^1)\sum_{s_j^2} P(s_j^2|s_1^1, m)\sum_{a'}\left[\frac{\sqrt{\pi^2(a'|s_j^2)\sigma^2(s_j^2, a')} + 4dc\sqrt{\frac{\log(H/\delta)}{b^*(a'|s_j^2)(T_L^K(s_j^2) - 2SA) + 1}}}{T_L^{*,K}(s_j^2, a')}\right]$$

$$
\overset{(c)}{\leq} \sum_{s_j^2} \left[ \frac{B(s_1^1)}{T_L^K(s_1^1)} + 4dc\frac{\sqrt{\log(H/\delta)}}{T_L^{(3/2),K}(s_1^1)b^*(m|s_1^1)^{3/2}} + \frac{4AB(s_1^1)}{T_L^{(2),K}(s_1^1)} \right]
$$

$$
+ \gamma\pi(m|s_1^1)\sum_{s_j^2} P(s_j^2|s_1^1,m) \underbrace{\sum_{a'} \left[ \frac{B(s_j^2)}{T_L^K(s_j^2)} + 4dc\frac{\sqrt{\log(H/\delta)}}{T_L^{(3/2),K}(s_j^2)b_{\min}^{*,3/2}(s_j^2)} + \frac{4AB(s_j^2)}{T_L^{(2),K}(s_j^2)-1} \right]}_{\mathbb{V}(s_j^2)}
$$

$$
\overset{(d)}{\leq} \sum_{s_j^2} \left[ \frac{B(s_1^1)}{T_L^K(s_1^1)} + 4dc\frac{\sqrt{\log(H/\delta)}}{T_L^{(3/2),K}(s_1^1)b_{\min}^{*,3/2}(s_1^1)} + \frac{4AB(s_1^1)}{T_L^{(2),K}(s_1^1)-1} \right] + \gamma\pi(a|s_1^1)\sum_{s_j^2} P(s_j^2|s_1^1,a)\mathbb{V}(s_j^2) \qquad (36)
$$

where, $(a)$ follows as $T_n^{*,(k)}(s_1^1,m) \geq T_L^K(s_1^1,m) - 1$ as action $m$ is over-pulled, $(b)$ follows as $T_L^K(s_1^1) = \sum_a T_L^K(s_1^1,a)$ and $T_n^k(s_1^1,m) - 2 > b^*(m|s_1^1)(T_L^K(s_1^1) - 2SA)$ and a similar argument follows in state $s_j^2$, $(c)$ follows $T_L^{*,K}(s_1^1,m) = \frac{\sqrt{\pi^2(m|s_1^1)\sigma^2(s_1^1,m)}}{B(s_1^1)}T_L^K(s_1^1)$, and using the result of lemma 6. Finally, $(d)$ follows as $b^*(m|s_1^1) \geq b_{\min}(s_1^1)$. In $(d)$ we also define the total over samples in state $s_j^2$ as $\mathbb{V}(s_j^2)$ such that

$$
\mathbb{V}(s_j^2) := \sum_{a'} \left[ \frac{B(s_j^2)}{T_L^K(s_j^2)} + 4dc\frac{\sqrt{\log(H/\delta)}}{T_L^{(3/2),K}(s_j^2)b_{\min}^{*,3/2}(s_j^2)} + \frac{4AB(s_j^2)}{T_L^{(2),K}(s_j^2)-1} \right]
$$

By rearranging (36) , we obtain the lower bound on $T_L^K(s_1^1,p)$ :

$$
T_L^K(s_1^1,p) \geq \frac{\sqrt{\pi^2(p|s_1^1)\sigma^2(s_1^1,p)}}{\frac{B(s_1^1)}{T_L^K(s_1^1)} + 4dc\frac{\sqrt{\log(H/\delta)}}{T_L^{(3/2),K}(s_1^1)b_{\min}^{*,3/2}(s_1^1)} + \frac{4AB(s_1^1)}{T_L^{(2),K}(s_1^1)} + \gamma\pi(m|s_1^1)\sum_{s_j^2} P(s_j^2|s_1^1,m)\mathbb{V}(s_j^2)}
$$

$$
= \frac{\sqrt{\pi^2(p|s_1^1)\sigma^2(s_1^1,p)}}{\frac{B(s_1^1)}{T_L^K(s_1^1)}} \left[ \frac{1}{1 + 4dc\frac{\sqrt{\log(H/\delta)}}{B(s_1^1)T_L^{(1/2),K}(s_1^1)b_{\min}^{*,3/2}(s_1^1)} + \frac{4A}{T_n^k(s_1^1)} + \gamma\pi(m|s_1^1)\frac{T_L^K(s_1^1)}{B(s_1^1)}\sum_{s_j^2} P(s_j^2|s_1^1,m)\mathbb{V}(s_j^2)} \right]
$$

$$
\geq \frac{\sqrt{\pi^2(p|s_1^1)\sigma^2(s_1^1,p)}}{\frac{B(s_1^1)}{T_L^K(s_1^1)}} \left[ \frac{1}{1 + 4dc\frac{\sqrt{\log(H/\delta)}}{B(s_1^1)T_L^{(1/2),K}(s_1^1)b_{\min}^{*,3/2}(s_1^1)} + \frac{4A}{T_n^k(s_1^1)} + \gamma\pi(m|s_1^1)\sum_{s_j^2} P(s_j^2|s_1^1,m)\mathbb{V}(s_j^2)} \right]
$$

$$
\overset{(a)}{\geq} \frac{\sqrt{\pi^2(p|s_1^1)\sigma^2(s_1^1,p)}}{\frac{B(s_1^1)}{T_L^K(s_1^1)}} \left[ 1 - 4dc\frac{\sqrt{\log(H/\delta)}}{B(s_1^1)T_L^{(1/2),K}(s_1^1)b_{\min}^{*,3/2}(s_1^1)} - \frac{4A}{T_n^k(s_1^1)} - \gamma\pi(m|s_1^1)\frac{T_L^K(s_1^1)}{B(s_1^1)}\sum_{s_j^2} P(s_j^2|s_1^1,m)\mathbb{V}(s_j^2) \right]
$$

$$
\overset{(b)}{=} \frac{\sqrt{\pi^2(p|s_1^1)\sigma^2(s_1^1,p)}}{\frac{B(s_1^1)}{T_L^K(s_1^1)}} \left[ 1 - 4dc\frac{\sqrt{\log(H/\delta)}}{B(s_1^1)T_L^{(1/2),K}(s_1^1)b_{\min}^{*,3/2}(s_1^1)} - \frac{4A}{T_n^k(s_1^1)} \right.
$$

$$
\left. - \gamma\pi(m|s_1^1)\frac{T_L^K(s_1^1)}{B(s_1^1)}\sum_{s_j^2} P(s_j^2|s_1^1,m)\left( \frac{B(s_j^2)}{T_L^K(s_j^2)} + 4dc\frac{\sqrt{\log(H/\delta)}}{T_L^{(3/2),K}(s_j^2)b_{\min}^{*,3/2}(s_j^2)} + \frac{4AB(s_j^2)}{T_L^{(2),K}(s_j^2)-1} \right) \right]
$$

$$\overset{(c)}{\geq} T_L^{*,K}(s_1^1,p) - 4dcb^*(p|s_1^1)\frac{\sqrt{\log(H/\delta)}}{B(s_1^1)b_{\min}^{*,3/2}(s_1^1)}\sqrt{T_L^K(s_1^1)} - 4Ab^*(p|s_1^1)$$

$$- \gamma\pi(m|s_1^1)\sum_{s_j^2}\frac{B(s_j^2)T_L^K(s_j^2)}{b^*(m|s_j^2)B(s_1^1)}P(s_j^2|s_1^1,m)\left(\frac{B(s_j^2)}{T_L^K(s_j^2)} + 4dc\frac{\sqrt{\log(H/\delta)}}{T_L^{(3/2),K}(s_j^2)b_{\min}^{*,3/2}(s_j^2)} + \frac{4AB(s_j^2)}{T_L^{(2),K}(s_j^2)-1}\right)\Bigg]$$

$$\geq T_L^{*,K}(s_1^1,p) - 4dcb^*(p|s_1^1)\frac{\sqrt{\log(H/\delta)}}{B(s_1^1)b_{\min}^{*,3/2}(s_1^1)}\sqrt{T_L^K(s_1^1)} - 4Ab^*(p|s_1^1)$$

$$- \gamma\pi(m|s_1^1)\frac{T_L^K(s_1^1)}{B^2(s_1^1)}\sum_{s_j^2}P(s_j^2|s_1^1,m)\left(\frac{B(s_j^2)}{b^*(m|s_j^2)}\sum_{a'}\left[T_L^{*,K}(s_j^2,a') + 4dcb^*(a'|s_j^2)\frac{\sqrt{\log(H/\delta)}}{b_{\min}^{*,3/2}(s_j^2)}\sqrt{T_L^K(s_1^1)} + 4Ab^*(a'|s_j^2)\right]\right)\Bigg]$$

$$\geq T_L^{*,K}(s_1^1,p) - 4dcb^*(p|s_1^1)\frac{\sqrt{\log(H/\delta)}}{B(s_1^1)b_{\min}^{*,3/2}(s_1^1)}\sqrt{T_L^K(s_1^1)} - 4Ab^*(p|s_1^1)$$

$$- \gamma\pi(m|s_1^1)\frac{T_L^K(s_1^1)}{B^2(s_1^1)}\sum_{s_j^2}P(s_j^2|s_1^1,m)\frac{B(s_j^2)}{b^*(m|s_j^2)}\sum_{a'}\left[T_L^{*,K}(s_j^2,a') + 4dcb^*(a'|s_j^2)\frac{\sqrt{\log(H/\delta)}}{b_{\min}^{*,3/2}(s_j^2)}\sqrt{T_L^K(s_1^1)} + 4Ab^*(a'|s_j^2)\right]$$

where in $(a)$ we use $1/(1+x) \geq 1-x$ (for $x > -1$ ), in $(b)$ we substitute the value $\mathbb{V}(s_j^2)$, and $(c)$ follows as $T_L^K(s_j^2) = \left(b(m|s_j^2)/B(s_j^2)\right)T_L^K(s_1^1)$. $\qquad\square$

**Lemma 8.** *Let the total budget be $n = KL$ and $n \geq 4SA$. Then the total regret in a deterministic $2$-depth $\mathbf{T}$ at the end of $K$-th episode when sampling according to the (8) is given by*

$$\mathcal{R}_n \leq \widetilde{O}\left(\frac{B^2(s_1^1)\sqrt{\log(SAn^{11/2})}}{n^{3/2}b_{\min}^{*,3/2}(s_1^1)} + \gamma\max_{s_j^2,a}\pi(a|s_1^1)P(s_j^2|s_1^1,a)\frac{B^2(s_j^2)\sqrt{\log(SAn^{11/2})}}{n^{3/2}b_{\min}^{*,3/2}(s_j^2)}\right)$$

*where, the $\widetilde{O}$ hides other lower order terms resulting out of the expansion of the squared terms and $B(s_i^\ell)$ is defined in (4).*

*Proof.* **Step 1 ($T_t^k(s_i^\ell,a)$ is a stopping time):** Let $\tau$ be a random variable, which is defined on the filtered probability space. Then $\tau$ is called a stopping time (with respect to the filtration $\left((\mathcal{F}_n)_{n\in\mathbb{N}}\right)$, if the following condition holds: $\{\tau = n\} \in \mathcal{F}_n$ for all $n$ Intuitively, this condition means that the "decision" of whether to stop at time $n$ must be based only on the information present at time $n$, not on any future information. Now consider the state $s_i^\ell$ and an action $a$. At each time step $t+1$, the ReVar algorithm decides which action to pull according to the current values of the upper-bounds $\left\{\widehat{\sigma^u}_{t+1}^k(s_i^\ell,a)\right\}_a$ in state $s_i^\ell$. Thus for any action $a$, $T_{t+1}^k(s_i^\ell,a)$ depends only on the values $\left\{T_{t+1}^k(s_i^\ell,a)\right\}_a$ and $\left\{\widehat{\sigma}_t^k(s_i^\ell,a)\right\}_k$ in state $s_i^\ell$. So by induction, $T_t^k(s_i^\ell,a)$ depends on the sequence of rewards $\left\{R_1^k(s_i^\ell,a),\ldots,R_{T_t^k(s_i^\ell,a)}^k(s_i^\ell,a)\right\}$, and on the samples of the other arms (which are independent of the samples of arm $k$ ). So we deduce that $T_L^K(s_i^\ell,a)$ is a stopping time adapted to the process $\left(R_t^k(s_i^\ell,a)\right)_{t\leq n}$.

**Step 2 (Regret bound):** By definition, given the dataset $\mathcal{D}$ after $K$ episodes each of trajectory length $L$, we have $n$ state-action samples. Then the loss of the algorithm is

$$\mathcal{L}_n = \mathbb{E}_{\mathcal{D}}\left[\left(Y_n(s_1^1) - v^\pi(s_1^1)\right)^2\right]$$
$$= \mathbb{E}_{\mathcal{D}}\left[\left(Y_n(s_1^1) - v^\pi(s_1^1)\right)^2\mathbb{I}\{\xi_\delta\}\right] + \mathbb{E}_{\mathcal{D}}\left[\left(Y_n(s_1^1) - v^\pi(s_1^1)\right)^2\mathbb{I}\left\{\xi_\delta^C\right\}\right]$$

where, $n = KL$ is the total budget. To handle the second term, we recall that $\xi_\delta^C$ holds with probability $2\delta$. Further due to the bounded reward assumption we have

$$\mathbb{E}_{\mathcal{D}}\left[\left(Y_n(s_1^1) - v^\pi(s_1^1)\right)^2\right] \leq 2n^2K\delta(4\eta^2+2\eta) \leq 2(4\eta^2+2\eta)n^2A\delta\left(1+\log\left(c_2/2nA\delta\right)\right)$$

where $c_2 > 0$ is a constant. Following Lemma 2 of [Carpentier and Munos, 2011] and setting $\delta = n^{-7/2}$ gives us an upper bounds of the quantity

$$\mathbb{E}_{\mathcal{D}}\left[\left(Y_n(s_1^1) - v^\pi(s_1^1)\right)^2\mathbb{I}\left\{\xi_\delta^C\right\}\right] \leq O\left(\frac{\log n}{n^{3/2}}\right).$$

Note that Carpentier and Munos [2011] uses a similar $\delta = n^{-7/2}$ due to the sub-Gaussian assumption on their reward distribution. Also observe that under the Assumption 2 we also have a sub-Gaussian assumption. Hence we can use Lemma 2 of Carpentier and Munos [2011]. Now, using the definition of $Y_n(s_1^1)$ and Lemma 2 we bound the first term as

$$
\mathbb{E}_{\mathcal{D}}\left[\left(Y_n(s_1^1) - v^\pi(s_1^1)\right)^2 \mathbb{I}\{\xi_\delta\}\right] \overset{(a)}{=} \mathbf{Var}[Y_n(s_1^1)]\mathbb{E}[T_L^K(s_1^1)]
$$

$$
\leq \sum_a \pi^2(a|s_1^1)\left[\frac{\sigma^2(s_1^1,a)}{\underline{T}_L^{(2),K}(s_1^1,a)}\right]\mathbb{E}[T_L^K(s_1^1,a)] + \gamma^2 \sum_a \pi^2(a|s_1^1)\sum_{s_j^2} P(s_j^2|s_1^1,a)\mathbf{Var}[Y_n(s_j^2)]\mathbb{E}[T_L^K(s_j^2,a)]
$$

$$
\leq \sum_a \pi^2(a|s_1^1)\left[\frac{\sigma^2(s_1^1,a)}{\underline{T}_L^{(2),K}(s_1^1,a)}\right]\mathbb{E}[T_L^K(s_1^1,a)] + \gamma^2 \sum_a \pi^2(a|s_1^1)\sum_{s_j^2} P(s_j^2|s_1^1,a)\sum_{a'} \pi^2(a'|s_j^2)\left[\frac{\sigma^2(s_j^2,a')}{\underline{T}_L^{(2),K}(s_j^2,a')}\right]\mathbb{E}[T_L^K(s_j^2,a')]
$$

(37)

where, $(a)$ follows from Lemma 2, and $\underline{T}_n(s_i^\ell,a)$ is the lower bound on $T_L^K(s_i^\ell,a)$ on the event $\xi_\delta$. Note that as $\sum_a T_L^K(s_1^1,a) = n$, we also have $\sum_a \mathbb{E}\left[T_L^K(s_1^1,a)\right] = n$. Using eq. (37) and eq. (36) for $\pi^2(a|s_1^1)\sigma^2(s_1^1,a)/\underline{T}_n^k(s_1^1,a)$ (which is equivalent to using a lower bound on $T_L^K(s_1^1,a)$ on the event $\xi_\delta$), we obtain

$$
\sum_a \pi^2(a|s_1^1)\left[\frac{\sigma^2(s_1^1,a)}{\underline{T}_L^{(2),K}(s_1^1,a)}\right]\mathbb{E}[T_n(s_1^1)] \leq \sum_a \left(\left[\frac{B(s_1^1)}{T_L^K(s_1^1)} + 4dc\frac{\sqrt{\log(H/\delta)}}{T_L^{(3/2),K}(s_1^1)b_{\min}^{*,3/2}(s_1^1)} + \frac{4AB(s_1^1)}{T_L^{(2),K}(s_1^1)-1}\right]\right.
$$

$$
\left. + \gamma\pi(a|s_1^1)\sum_{s_j^2} P(s_j^2|s_1^1,a)\sum_{a'}\left[\frac{B(s_j^2)}{T_L^K(s_j^2)} + 4dc\frac{\sqrt{\log(H/\delta)}}{T_L^{(3/2),K}(s_j^2)b_{\min}^{*,3/2}(s_j^2)} + \frac{4AB(s_j^2)}{T_L^{(2),K}(s_j^2)-1}\right]\right)^2 \mathbb{E}[T_L^K(s_1^1,a)].
$$

(38)

Finally the R.H.S. of eq. (38) may be bounded using the fact that $\sum_a \mathbb{E}\left[T_L^K(s_1^1,a)\right] = n$ as

$$
\sum_a \pi^2(a|s_1^1)\left[\frac{\sigma^2(s_1^1,a)}{\underline{T}_L^{(2),K}(s_1^1,a)}\right]\mathbb{E}[T_L^K(s_1^1)] \leq \sum_a \left(\left[\frac{B(s_1^1)}{T_L^K(s_1^1)} + 4dc\frac{\sqrt{\log(H/\delta)}}{T_L^{(3/2),K}(s_1^1)b_{\min}^{*,3/2}(s_1^1)} + \frac{4AB(s_1^1)}{T_L^{(2),K}(s_1^1)-1}\right]\right.
$$

$$
\left. + \gamma\pi(a|s_1^1)\sum_{s_j^2} P(s_j^2|s_1^1,a)\sum_{a'}\left[\frac{B(s_j^2)}{T_L^K(s_j^2)} + 4dc\frac{\sqrt{\log(H/\delta)}}{T_L^{(3/2),K}(s_j^2)b_{\min}^{*,3/2}(s_j^2)} + \frac{4AB(s_j^2)}{T_K^{(2),L}(s_j^2)-1}\right]\right)^2 \mathbb{E}[T_L^K(s_1^1,a)]
$$

$$
\overset{(a)}{\leq} 2\left(\left[\frac{B(s_1^1)}{T_L^K(s_1^1)} + 4dc\frac{\sqrt{\log(H/\delta)}}{T_L^{(3/2),K}(s_1^1)b_{\min}^{*,3/2}(s_1^1)} + \frac{4AB(s_1^1)}{T_L^{(2),K}(s_1^1)-1}\right]\right)^2 \sum_a \mathbb{E}[T_L^K(s_1^1,a)]
$$

$$
+ 2\left(\gamma\pi(a|s_1^1)\sum_{s_j^2} P(s_j^2|s_1^1,a)\sum_{a'}\left[\frac{B(s_j^2)}{T_L^K(s_j^2)} + 4dc\frac{\sqrt{\log(H/\delta)}}{T_L^{(3/2),K}(s_j^2)b_{\min}^{*,3/2}(s_j^2)} + \frac{4AB(s_j^2)}{T_L^{(2),K}(s_j^2)-1}\right]\right)^2 \sum_a \mathbb{E}[T_L^K(s_1^1,a)]
$$

$$
\overset{(b)}{\leq} \widetilde{O}\left(\frac{B^2(s_1^1)\sqrt{\log(H/\delta)}}{n^{3/2}b_{\min}^{*,3/2}(s_1^1)} + \gamma\max_{s_j^2,a}\pi(a|s_1^1)P(s_j^2|s_1^1,a)\frac{B^2(s_j^2)\sqrt{\log(H/\delta)}}{n^{3/2}b_{\min}^{*,3/2}(s_j^2)}\right)
$$

$$
\overset{(c)}{=} \widetilde{O}\left(\frac{B^2(s_1^1)\sqrt{\log(SAn^{11/2})}}{n^{3/2}b_{\min}^{*,3/2}(s_1^1)} + \gamma\max_{s_j^2,a}\pi(a|s_1^1)P(s_j^2|s_1^1,a)\frac{B^2(s_j^2)\sqrt{\log(SAn^{11/2})}}{n^{3/2}b_{\min}^{*,3/2}(s_j^2)}\right)
$$

where, $(a)$ follows as $(a+b)^2 \leq 2(a^2+b^2)$ for any $a,b > 0$, in $(b)$ we have $T_L^K(s_1^1) = n$, and the $\widetilde{O}$ hides other lower order terms resulting out of the expansion of the squared terms, and $(c)$ follows by setting $\delta = n^{-7/2}$ and using $H = SAn(n+1)$. $\qquad\square$

## H   REGRET FOR A DETERMINISTIC $L$-DEPTH TREE

**Theorem 2.** *Let the total budget be $n = KL$ and $n \geq 4SA$. Then the total regret in a deterministic L-depth $\mathbf{T}$ at the end of K-th episode when taking actions according to (8) is given by*

$$
\mathcal{R}_n \leq \widetilde{O}\left(\frac{B_{s_1^1}^2\sqrt{\log(SAn^{11/2})}}{n^{3/2}b_{\min}^{*,3/2}(s_1^1)} + \gamma\sum_{\ell=2}^L \max_{s_j^\ell,a}\pi(a|s_1^1)P(s_j^\ell|s_1^1,a)\frac{B_{s_j^\ell}^2\sqrt{\log(SAn^{11/2})}}{n^{3/2}b_{\min}^{*,3/2}(s_j^\ell)}\right)
$$

*where, the $\widetilde{O}$ hides other lower order terms and $B_{s_i^\ell}$ is defined in (4) and $b_{\min}^*(s) = \min_a b^*(a|s)$.*

**Proof.** The proof follows directly by using Lemma 6, Lemma 7, and Lemma 8.

**Step 1 ($T_t^k(s_i^\ell, a)$ is a stopping time):** This step is same as Lemma 8 as all the arguments hold true even for the $L$ depth deterministic tree.

**Step 2 (MSE decomposition):** Given the dataset $\mathcal{D}$ of $K$ episodes each of trajectory length $L$, the MSE of the algorithm is

$$\mathcal{L}_n = \mathbb{E}_{\mathcal{D}}\left[\left(Y_n(s_1^1) - v^\pi(s_1^1)\right)^2\right] = \mathbb{E}_{\mathcal{D}}\left[\left(Y_n(s_1^1) - v^\pi(s_1^1)\right)^2 \mathbb{I}\{\xi_\delta\}\right] + \mathbb{E}_{\mathcal{D}}\left[\left(Y_n(s_1^1) - v^\pi(s_1^1)\right)^2 \mathbb{I}\{\xi_\delta^C\}\right]$$

where, $n = KL$ is the total budget. Using Lemma 8 we can upper bound the second term as $O\left(n^{-3/2}\log(n)\right)$. Using the definition of $Y_n(s_1^1)$ and Lemma 2 we bound the first term as

$$
\begin{aligned}
\mathbb{E}_{\mathcal{D}}\left[\left(Y_n(s_1^1) - v^\pi(s_1^1)\right)^2 \mathbb{I}\{\xi_\delta\}\right] &\overset{(a)}{=} \mathbf{Var}[Y_n(s_1^1)]\mathbb{E}[T_L^K(s_1^1)] = \\
&\overset{b}{\leq} \sum_a \pi^2(a|s_1^1)\left[\frac{\sigma^2(s_1^1, a)}{\underline{T}_L^{(2),K}(s_1^1, a)}\right]\mathbb{E}[T_L^K(s_1^1, a)] + \gamma^2 \sum_a \pi^2(a|s_1^1)\sum_{s_j^2} P(s_j^2|s_1^1, a)\mathbf{Var}[Y_n(s_j^2)]\mathbb{E}[T_L^K(s_j^2)] \\
&\overset{(c)}{\leq} \sum_a \pi^2(a|s_1^1)\left[\frac{\sigma^2(s_1^1, a)}{\underline{T}_L^{(2),K}(s_1^1, a)}\right]\mathbb{E}[T_L^K(s_1^1, a)] \\
&\quad + \gamma^2 \sum_a \pi^2(a|s_1^1)\sum_{\ell=2}^L \sum_{s_j^\ell} P(s_j^\ell|s_1^1, a) \sum_{a'} \pi^2(a'|s_j^\ell)\left[\frac{\sigma^2(s_j^\ell, a')}{\underline{T}_L^{(2),K}(s_j^\ell, a')}\right]\mathbb{E}[T_L^K(s_j^\ell, a')]
\end{aligned}
\tag{39}
$$

where, $(a)$ follows from Lemma 2, $(b)$ follows from by unrolling the variance for $Y_n(s_1^1)$, and where $\underline{T}_n(s_i^\ell, a)$ is the lower bound on $T_L^K(s_i^\ell, a)$ on the event $\xi_\delta$. Finally, $(c)$ follows by unrolling the variance for all the states till level $L$ and taking the lower bound of $\underline{T}_n(s_i^\ell, a)$ for each state-action pair.

**Step 2 (MSE at level $L$):** Now we want to upper bound the total MSE in (39). Using eq. (33) in Lemma 6 we can directly get the MSE upper bound for a state $s_i^L$ as

$$\sum_{a'} \pi^2(a'|s_i^L)\left[\frac{\sigma^2(s_i^L, a')}{\underline{T}_L^{(2),K}(s_i^L, a')}\right]\mathbb{E}[T_L^K(s_i^L, a')] \leq \widetilde{O}\left(\frac{B_{s_i^L}^2 \sqrt{\log(SAn^{11/2})}}{n^{3/2}b_{\min}(s_i^L)}\right).$$

**Step 3 (MSE at level $L-1$):** This step follows directly from eq. (36) in Lemma 7. We can get the loss upper bound for a state $s_i^{L-1}$ (which takes into account the loss at level $L$ as well) as follows:

$$\sum_{a'} \left(\frac{b^*(a'|s_i^{L-1})}{\underline{T}_L^{(2),K}(s_i^{L-1}, a')}\right)\mathbb{E}[T_L^K(s_i^{L-1}, a')] \leq \widetilde{O}\left(\frac{B_{s_i^{L-1}}^2 \sqrt{\log(SAn^{11/2})}}{n^{3/2}b_{\min}(s_i^{L-1})} + \gamma \max_{s_j^L, a} \pi(a|s_i^{L-1})P(s_j^L|s_i^{L-1}, a)\frac{B_{s_j^L}^2 \sqrt{\log(SAn^{11/2})}}{n^{3/2}b_{\min}(s_j^L)}\right).$$

**Step 4 (MSE at arbitrary level $\ell$):** This step follows by combining the results of step 2 and 3 iteratively from states in level $\ell$ to $L$ under the good event $\xi_\delta$. We can get the regret upper bound for a state $s_i^\ell$ as

$$\sum_{a'} \left(\frac{b^*(a'|s_i^\ell)}{\underline{T}_L^{(2),K}(s_i^\ell, a')}\right)\mathbb{E}[T_L^K(s_i^\ell, a')] \leq \widetilde{O}\left(\frac{B_{s_i^\ell}^2 \sqrt{\log(SAn^{11/2})}}{n^{3/2}b_{\min}(s_i^\ell)} + \gamma \sum_{\ell'=\ell+1}^L \max_{s_j^{\ell'}, a} \pi(a|s_i^{\ell'-1})P(s_j^{\ell'}|s_i^{\ell'-1}, a)\frac{B_{s_j^{\ell'}}^2 \sqrt{\log(SAn^{11/2})}}{n^{3/2}b_{\min}(s_j^{\ell'})}\right).$$

**Step 4 (Regret at level 1):** Finally, combining all the steps above we get the regret upper bound for the state $s_1^1$ as follows

$$\mathcal{R}_n = \mathcal{L}_n - \mathcal{L}_n^* = \widetilde{O}\left(\frac{B^2(s_1^1)\sqrt{\log(SAn^{11/2})}}{n^{3/2}b_{\min}^{*,3/2}(s_1^1)} + \gamma \sum_{\ell=2}^L \max_{s_j^\ell, a} \pi(a|s_1^1)P(s_j^\ell|s_1^1, a)\frac{B^2(s_j^\ell)\sqrt{\log(SAn^{11/2})}}{n^{3/2}b_{\min}^{*,3/2}(s_j^\ell)}\right).$$

$\square$

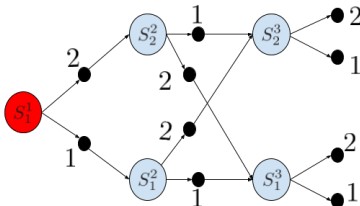

Figure 5: A 3-depth 2-Action DAG

**Remark 2. (Stochastic MDP extension):** Observe that the Theorem 2 is quite general as the regret

$$\mathcal{R}_n \leq \tilde{O}\left(\frac{B_{s_1^1}^2\sqrt{\log(SAn^{11/2})}}{n^{3/2}b_{\min}^{*,3/2}(s_1^1)} + \gamma\sum_{\ell=2}^{L}\max_{s_j^\ell,a}\pi(a|s_1^1)P(s_j^\ell|s_1^1,a)\frac{B_{s_j^\ell}^2\sqrt{\log(SAn^{11/2})}}{n^{3/2}b_{\min}^{*,3/2}(s_j^\ell)}\right)$$

incorporates the transition probability $P(s'|s,a)$. Hence, the result of Theorem 2 holds not only for the deterministic case but also for the stochastic setting, when the algorithm is provided with the knowledge of $P(s'|s,a)$ upto some constant scaling. Note that ReVar does not perform any exploration to estimate the transition probabilities, and it is not clear how to extend the current UCB based approach that minimizes MSE to also estimate the $P$. We leave this direction for future works.

## I  DAG OPTIMAL SAMPLING

**Proposition 3.** *(Restatement)* *Let $\mathcal{G}$ be a $3$-depth, $A$-action DAG defined in Definition 2. The minimal-MSE sampling proportions $b^*(a|s_1^1), b^*(a|s_j^2)$ depend on themselves such that $b(a|s_1^1) \propto f(1/b(a|s_1^1))$ and $b(a|s_j^2) \propto f(1/b(a|s_j^2))$ where $f(\cdot)$ is a function that hides other dependencies on variances of $s$ and its children.*

*Proof.* **Step 1 (Level** 3**):** For an arbitrary state $s_i^3$ we can calculate the expectation and variance of $Y_n(s_i^3)$ as follows:

$$\mathbb{E}[Y_n(s_i^3)] = \sum_a \frac{\pi(a|s_i^3)}{T_n(s_i^3,a)}\sum_{h=1}^{T_n(s_i^3,a)}\mathbb{E}[R_h(s_i^3,a)] = \sum_a \pi(a|s_i^3)\mu(s_i^3,a)$$

$$\mathbf{Var}[Y_n(s_i^3)] = \sum_a \frac{\pi^2(a|s_i^3)}{T_n^2(s_i^3,a)}\sum_{h=1}^{T_n(s_i^3,a)}\mathbf{Var}[R_h(s_i^3,a)] = \sum_a \frac{\pi^2(a|s_i^3)}{T_n(s_i^3,a)}\sigma^2(s_i^3,a).$$

**Step 2 (Level** 2**):** For the arbitrary state $s_i^2$ we can calculate the expectation of $Y_n(s_1^2)$ as follows:

$$\mathbb{E}[Y_n(s_i^2)] = \sum_a \frac{\pi(a|s_i^2)}{T_n(s_i^2,a)}\sum_{h=1}^{T_n(s_i^2,a)}\mathbb{E}[R_h(s_i^2,a)] + \gamma\sum_a\pi(a|s_i^2)\sum_{s_j^3}P(s_j^3|s_i^2,a)\sum_{a'}\frac{\pi(a'|s_j^3)}{T_n(s_j^3,a')}\sum_{h=1}^{T_n(s_j^3,a')}\mathbb{E}[R_h(s_j^3,a')]$$

$$= \sum_a\pi(a|s_i^2)\left(\mu(s_i^2,a) + \gamma\sum_{s_j^3}P(s_j^3|s_i^2,a)\mathbb{E}[Y_n(s_j^3)]\right)$$

$$\mathbf{Var}[Y_n(s_i^2)] = \sum_a \frac{\pi^2(a|s_i^2)}{T_n^2(s_1^2,a)}\sum_{h=1}^{T_n(s_1^2,a)}\mathbf{Var}[R_h(s_i^2,a)] + \gamma^2\sum_a\pi^2(a|s_1^2)\sum_{s_j^3}P(s_j^3|s_i^2,a)\sum_{a'}\frac{\pi^2(a'|s_j^3)}{T_n^2(s_j^3,a')}\sum_{h=1}^{T_n(s_j^3,a')}\mathbf{Var}[R_h(s_j^3,a')]$$

$$= \sum_a \frac{\pi^2(a|s_i^2)}{T_n(s_1^2,a)}\left(\sigma^2(s_1^2,a) + \gamma^2\sum_{s_j^3}P(s_j^3|s_1^2,a)\mathbf{Var}[Y_n(s_j^3)]\right)$$

**Step 3 (Level 1):** Finally for the state $s_1^1$ we can calculate the expectation and variance of $Y_n(s_1^1)$ as follows:

$$\mathbb{E}[Y_n(s_1^1)] = \sum_a \frac{\pi(a|s_1^1)}{T_n(s_1^1,a)} \sum_{h=1}^{T_n(s_1^1,a)} \mathbb{E}[R_h(s_1^1,a)] + \gamma \pi(a|s_1^1) \sum_{s_j^2} P(s_j^2|s_1^1,a) \sum_{a'} \frac{\pi(a'|s_j^2)}{T_n(s_j^2,a')} \sum_{h=1}^{T_n(s_j^2,a')} \mathbb{E}[R_h(s_j^2,a')]$$

$$= \sum_a \pi(a|s_1^1) \left( \mu(s_1^1,a) + \gamma \sum_{s_j^2} P(s_j^2|s_1^1,a) \mathbb{E}[Y_n(s_j^2)] \right)$$

$$\mathbf{Var}[Y_n(s_1^1)] = \sum_a \frac{\pi^2(a|s_1^1)}{T_n^2(s_1^1,a)} \sum_{h=1}^{T_n(s_1^1,a)} \mathbf{Var}[R_h(s_1^1,a)] + \gamma^2 \sum_a \pi^2(a|s_1^1) \sum_{s_j^2} P(s_j^2|s_1^1,a) \sum_{a'} \frac{\pi^2(a'|s_j^2)}{T_n^2(s_j^2,a')} \sum_{h=1}^{T_n(s_j^2,a')} \mathbf{Var}[R_h(s_j^2,a')]$$

$$= \sum_a \frac{\pi^2(a|s_1^1)}{T_n(s_1^1,a)} \left( \sigma^2(s_1^1,a) + \gamma^2 \sum_{s_j^2} P(s_j^2|s_1^1,a) \mathbf{Var}[Y_n(s_j^2)] \right)$$

Unrolling out the above equation we re-write the equation below:

$$\mathbf{Var}[Y_n(s_1^1)] = \sum_a \frac{\pi^2(a|s_1^1)\sigma^2(s_1^1,a)}{T_n(s_1^1,a)} + \sum_a \pi^2(a|s_1^1) \sum_{s_j^2} \sum_{a'} \frac{\pi^2(a'|s_j^2)\sigma^2(s_j^2,a')}{T_n(s_j^2,a')}$$

$$+ \sum_a \pi^2(a|s_1^1) \sum_{s_j^2} \sum_{a'} \pi^2(a'|s_j^2) \sum_{s_m^3} \sum_{a''} \frac{\pi^2(a''|s_m^3)\sigma^2(s_m^3,a'')}{T_n(s_m^3,a'')} \tag{40}$$

Since we follow a path $s_1^1 \xrightarrow{a} s_j^2 \xrightarrow{a'} s_m^3 \xrightarrow{a''}$ Terminate for any $a, a', a'' \in \mathcal{A}$ and $j, m \in \{1, 2, \ldots, A\}$. Hence we have the following constraints

$$\sum_a T_n(s_1^1,a) = n \tag{41}$$

$$\sum_a T_n(s_i^2,a) \overset{(a)}{=} \sum_a P(s_i^2|s_1^1,a)T_n(s_1^1,a) \tag{42}$$

$$\sum_a T_n(s_i^3,a) \overset{(b)}{=} \sum_{s_j^2} \sum_{a'} P(s_i^3|s_j^2,a')T_n(s_j^2,a') \tag{43}$$

observe that in $(a)$ in the deterministic case the $\sum_a P(s_i^2|s_1^1,a)T_n(s_1^1,a)$ is all the possible paths from $s_1^1$ to $s_i^2$ that were taken for $n$ samples over any action $a$. Similarly in $(b)$ in the deterministic case the $\sum_{a'} P(s_i^3|s_j^2,a)T_n(s_j^2,a')$ is all the possible paths from $s_j^2$ to $s_i^3$ that were taken for $n$ samples over any action $a'$.

**Step 4 (Formulate objective):** We want to minimize the variance in (40) subject to the above constraints. We can show that

$$T_n(s_1^1,a)/n = b(a|s_1^1). \tag{44}$$

$$\text{and,} \quad b(a|s_i^2) = \frac{T_n(s_i^2,a)}{\sum_{a'} T_n(s_i^2,a')} = \frac{T_n(s_i^2,a)}{\sum_{a'} P(s_i^2|s_1^1,a')T_n(s_1^1,a')} \overset{(a)}{=} \frac{T_n(s_i^2,a)/n}{\sum_{a'} P(s_i^2|s_1^1,a')T_n(s_1^1,a')/n}$$

$$\implies T_n(s_i^2,a)/n \overset{(b)}{=} b(a|s_i^2) \sum_{a'} P(s_i^2|s_1^1,a')b(a'|s_1^1), \tag{45}$$

where, $(a)$ follows from (42), and $(b)$ follows from (44) and taking into account all the possible paths to reach $s_i^2$ from $s_1^1$.

For the third level we can show that the proportion

$$b(a|s_i^3) = \frac{T_n(s_i^3, a)}{\sum_{a'} T_n(s_i^3, a')} \stackrel{(a)}{=} \frac{T_n(s_i^3, a)}{\sum_{s_j^2} \sum_{a'} P(s_i^3|s_j^2, a') T_n(s_j^2, a')}$$

$$\stackrel{(b)}{=} \frac{T_n(s_i^3, a)}{\sum_{s_j^2} \sum_{a'} P(s_j^2|s_1^1, a') b(a'|s_1^1) \sum_{a''} P(s_i^3|s_j^2, a'') b(a''|s_j^2)}$$

$$\implies T_n(s_i^3, a)/n = b(a|s_i^3) \sum_{s_j^2} \sum_{a'} P(s_j^2|s_1^1, a') b(a'|s_1^1) \sum_{a''} P(s_i^3|s_j^2, a'') b(a''|s_j^2)$$

where, $(a)$ follows from (43), and $(b)$ follows from (44) and taking into account all the possible paths to reach $s_i^3$ from $s_1^1$. Again note that we use $b(a|s)$ to denote the optimization variable and $b^*(a|s)$ to denote the optimal sampling proportion. Then the optimization problem in (40) can be restated as,

$$\min_{\mathbf{b}} \sum_a \frac{\pi^2(a|s_1^1)\sigma^2(s_1^1, a)}{b(a|s_1^1)} + \sum_a \pi^2(a|s_1^1) \sum_{s_j^2} \sum_{a'} \frac{\pi^2(a'|s_j^2)\sigma^2(s_j^2, a')}{b(a'|s_j^2) \underbrace{\sum_{a_1} P(s_j^2|s_1^1, a_1) b(a_1|s_1^1)}_{\text{All possible path to reach } s_j^2 \text{ from } s_1^1}}$$

$$+ \sum_a \pi^2(a|s_1^1) \sum_{s_j^2} \sum_{a'} \pi^2(a'|s_j^2) \sum_{s_m^3} \sum_{a''} \frac{\pi^2(a''|s_m^3)\sigma^2(s_m^3, a'')}{b(a''|s_m^3) \underbrace{\sum_{s_j^2} \sum_{a_1} P(s_j^2|s_1^1, a_1) b(a_1|s_1^1) \sum_{a_2} P(s_i^3|s_j^2, a_2) b(a_2|s_j^2)}_{\text{All possible path to reach } s_m^3 \text{ from } s_1^1}}$$

**s.t.** $\forall s, \quad \sum_a b(a|s) = 1$

$\forall s, a \quad b(a|s) > 0.$

Now introducing the Lagrange multiplier we get that

$$L(\mathbf{b}, \lambda) = \min_{\mathbf{b}} \sum_a \frac{\pi^2(a|s_1^1)\sigma^2(s_1^1, a)}{b(a|s_1^1)} + \sum_a \pi^2(a|s_1^1) \sum_{s_j^2} \sum_{a'} \frac{\pi^2(a'|s_j^2)\sigma^2(s_j^2, a')}{b(a'|s_j^2) \sum_{a_1} P(s_j^2|s_1^1, a_1) b(a_1|s_1^1)}$$

$$+ \sum_a \pi^2(a|s_1^1) \sum_{s_j^2} \sum_{a'} \pi^2(a'|s_j^2) \sum_{s_m^3} \sum_{a''} \frac{\pi^2(a''|s_m^3)\sigma^2(s_m^3, a'')}{b(a''|s_m^3) \sum_{s_j^2} \sum_{a_1} P(s_j^2|s_1^1, a_1) b(a_1|s_1^1) \sum_{a_2} P(s_i^3|s_j^2, a_2) b(a_2|s_j^2)}$$

$$+ \sum_s \lambda_s \left( \sum_a b(a|s) - 1 \right). \tag{46}$$

Now we need to solve for the KKT condition. Differentiating (46) with respect to $b(a''|s_m^3)$, $b(a'|s_j^2)$, $b(a|s_1^1)$, and $\lambda_s$ we get

$$\nabla_{b(a''|s^3_m)}L(\mathbf{b},\lambda) = -\sum_a \pi^2(a|s^1_1)\sum_{s^2_j}\sum_{a'}\pi^2(a'|s^2_j)\sum_{s^3_m}\sum_{a''}\frac{\pi^2(a''|s^3_m)\sigma^2(s^3_m,a'')}{b^2(a''|s^3_m)\sum_{s^2_j}\sum_{a_1}P(s^2_j|s^1_1,a_1)b(a_1|s^1_1)\sum_{a_2}P(s^3_i|s^2_j,a_2)b(a_2|s^2_j)}$$
$$+ \lambda_{s^3_m} \tag{47}$$

$$\nabla_{b(a'|s^2_j)}L(\mathbf{b},\lambda) = -\sum_a \pi^2(a|s^1_1)\sum_{s^2_j}\sum_{a'}\frac{\pi^2(a'|s^2_j)\sigma^2(s^2_j,a')}{b^2(a'|s^2_j)\sum_{a_1}P(s^2_j|s^1_1,a_1)b(a_1|s^1_1)} \tag{48}$$

$$- \sum_a \pi^2(a|s^1_1)\sum_{s^2_j}\sum_{a'}\pi^2(a'|s^2_j)\sum_{s^3_m}\sum_{a''}\frac{\pi^2(a''|s^3_m)\sigma^2(s^3_m,a'')}{b(a''|s^3_m)\left(\sum_{s^2_j}\sum_{a_1}P(s^2_j|s^1_1,a_1)b(a_1|s^1_1)\sum_{a_2}P(s^3_i|s^2_j,a_2)b(a_2|s^2_j)\right)^2}$$
$$+ \lambda_{s^2_j}$$

$$\nabla_{b(a|s^1_1)}L(\mathbf{b},\lambda) = -\sum_a \frac{\pi^2(a|s^1_1)\sigma^2(s^1_1,a)}{b^2(a|s^1_1)} - \sum_a \pi^2(a|s^1_1)\sum_{s^2_j}\sum_{a'}\frac{\pi^2(a'|s^2_j)\sigma^2(s^2_j,a')}{b(a'|s^2_j)\left(\sum_{a_1}P(s^2_j|s^1_1,a_1)b(a_1|s^1_1)\right)^2} \tag{49}$$

$$- \sum_a \pi^2(a|s^1_1)\sum_{s^2_j}\sum_{a'}\pi^2(a'|s^2_j)\sum_{s^3_m}\sum_{a''}\frac{\pi^2(a''|s^3_m)\sigma^2(s^3_m,a'')}{b(a''|s^3_m)\left(\sum_{s^2_j}\sum_{a_1}P(s^2_j|s^1_1,a_1)b(a_1|s^1_1)\sum_{a_2}P(s^3_i|s^2_j,a_2)b(a_2|s^2_j)\right)^2}$$
$$+ \lambda_{s^1_1}$$

$$\nabla_{\lambda_s}L(\mathbf{b},\lambda) = \sum_a b(a|s) - 1. \tag{50}$$

Now to remove $\lambda_{s^3_m}$ from (47) we first set (47) to 0 and show that

$$\lambda_{s^3_m} = \sum_a \pi^2(a|s^1_1)\sum_{s^2_j}\sum_{a'}\pi^2(a'|s^2_j)\sum_{s^3_m}\sum_{a''}\frac{\pi^2(a''|s^3_m)\sigma^2(s^3_m,a'')}{b^2(a''|s^3_m)\left(\sum_{s^2_j}\sum_{a_1}P(s^2_j|s^1_1,a_1)b(a_1|s^1_1)\sum_{a_2}P(s^3_i|s^2_j,a_2)b(a_2|s^2_j)\right)^2}$$

$$\implies b(a''|s^3_m) = \sqrt{\frac{1}{\lambda_{s^3_m}}\sum_a \pi^2(a|s^1_1)\sum_{s^2_j}\sum_{a'}\pi^2(a'|s^2_j)\sum_{s^3_m}\sum_{a''}\frac{\pi^2(a''|s^3_m)\sigma^2(s^3_m,a'')}{\left(\sum_{s^2_j}\sum_{a_1}P(s^2_j|s^1_1,a_1)b(a_1|s^1_1)\sum_{a_2}P(s^3_i|s^2_j,a_2)b(a_2|s^2_j)\right)^2}}. \tag{51}$$

Then setting (50) to 0 we have

$$\sum_{a''}\sqrt{\frac{1}{\lambda_{s^3_m}}\sum_a \pi^2(a|s^1_1)\sum_{s^2_j}\sum_{a'}\pi^2(a'|s^2_j)\sum_{s^3_m}\sum_{a''}\frac{\pi^2(a''|s^3_m)\sigma^2(s^3_m,a'')}{\left(\sum_{s^2_j}\sum_{a_1}P(s^2_j|s^1_1,a_1)b(a_1|s^1_1)\sum_{a_2}P(s^3_i|s^2_j,a_2)b(a_2|s^2_j)\right)^2}} = 1$$

$$\implies \lambda_{s^3_m} = \sum_{a''}\sqrt{\sum_a \pi^2(a|s^1_1)\sum_{s^2_j}\sum_{a'}\pi^2(a'|s^2_j)\sum_{s^3_m}\sum_{a''}\frac{\pi^2(a''|s^3_m)\sigma^2(s^3_m,a'')}{\left(\sum_{s^2_j}\sum_{a_1}P(s^2_j|s^1_1,a_1)b(a_1|s^1_1)\sum_{a_2}P(s^3_i|s^2_j,a_2)b(a_2|s^2_j)\right)^2}} \tag{52}$$

Using (51) and (52) we can show that the optimal sampling proportion is given by

$$b^*(a''|s^3_m) = \frac{\pi^2(a''|s^3_m)\sigma^2(s^3_m,a'')}{\sum_a \pi^2(a|s^3_m)\sigma^2(s^3_m,a)}$$

Similarly we can show that setting (48) and (50) setting to 0 and removing $\lambda_{s_j^2}$

$$b^{*,(2)}(a'|s_j^2) \propto \sum_a \pi^2(a|s_1^1) \sum_{s_j^2} \sum_{a'} \frac{\pi^2(a'|s_j^2)\sigma^2(s_j^2,a')}{\sum_{a_1} P(s_j^2|s_1^1,a_1)b^*(a_1|s_1^1)}$$

$$+ \sum_a \pi^2(a|s_1^1) \sum_{s_j^2} \sum_{a'} \pi^2(a'|s_j^2) \sum_{s_m^3} \sum_{a''} \frac{\pi^2(a''|s_m^3)\sigma^2(s_m^3,a'')}{b^*(a''|s_m^3)\left(\frac{\sum_{s_j^2}\sum_{a_1}P(s_j^2|s_1^1,a_1)b^*(a_1|s_1^1)\sum_{a_2}P(s_i^3|s_j^2,a_2)b^*(a_2|s_j^2)}{b^*(a'|s_j^2)}\right)^2}$$

Finally, setting (49) and (50) setting to 0 and removing $\lambda_{s_1^1}$ we have

$$b^{*,(2)}(a|s_1^1) \propto \sum_a \pi^2(a|s_1^1)\sigma^2(s_1^1,a) + \sum_a \pi^2(a|s_1^1) \sum_{s_j^2} \sum_{a'} \frac{\pi^2(a'|s_j^2)\sigma^2(s_j^2,a')}{b^*(a'|s_j^2)}$$

$$+ \sum_a \pi^2(a|s_1^1) \sum_{s_j^2} \sum_{a'} \pi^2(a'|s_j^2) \sum_{s_m^3} \sum_{a''} \frac{\pi^2(a''|s_m^3)\sigma^2(s_m^3,a'')}{b^*(a''|s_m^3)\left(\frac{\sum_{s_j^2}\sum_{a_1}P(s_j^2|s_1^1,a_1)b^*(a_1|s_1^1)\sum_{a_2}P(s_i^3|s_j^2,a_2)b^*(a_2|s_j^2)}{b^*(a|s_1^1)}\right)^2}$$

This shows the cyclical dependency of $b^*(a|s_1^1)$ and $b^*(a|s_j^2)$.  □

# J  ADDITIONAL EXPERIMENTAL DETAILS

## J.1  ESTIMATE $B$ IN DAG

Recall that in a DAG $\mathcal{G}$ we have a cyclical dependency following Proposition 3. Hence, we do an approximation of the optimal sampling proportion in $\mathcal{G}$ by using the tree formulation from Theorem 1. However, since there are multiple paths to the same state in $\mathcal{G}$ we have to iteratively compute the normalization factor $B$. To do this we use the following Algorithm 2.

---

**Algorithm 2** Estimate $B_0(s)$ for $\mathcal{G}$

---

1: Initialize $B_L(s) = 0$ for all $s \in \mathcal{S}$
2: **for** $t' \in L-1, \ldots, 0$ **do**

3: $\qquad B_{t'}(s) = \sum_a \sqrt{\pi^2(a|s)\left(\sigma^2(s,a) + \gamma^2 \sum_{s'} P(s'|s,a)B_{t'+1}^2(s)\right)}$

4: **Return** $B_0$.

---

## J.2  IMPLEMENTATION DETAILS

In this section we state additional experimental details. We implement the following competitive baselines:

**(1)** Onpolicy: The Onpolicy baseline follows the target probability when sampling actions at each state.

**(2)** CB-Var: This baseline is a bandit policy which samples an action based only on the statistics of the current state. At every time $t+1$ in episode $k$, CB-Var sample an action

$$I_{t+1}^k = \underset{a \in \mathcal{A}}{\arg\max}(2\eta + 4\eta^2)\sqrt{\frac{2\pi(a|s)\widehat{\sigma}_t^{(2),k}(s,a)\log(SAn(n+1))}{T_t^k(s,a)}} + \frac{7\log(SAn(n+1))}{3T_t^k(s,a)}$$

where, $n$ is the total budget. This policy is similar to UCB-variance of Audibert et al. [2009] and uses the empirical Bernstein inequality [Maurer and Pontil, 2009]. However we do not use the mean estimate $\widehat{\mu}_t^k(s,a)$ of an action so that CB-Var explores continuously rather than maximizing the rewards. Also note that to have a fair comparison with ReVar we use a large constant $(2\eta + 4\eta^2)$ and $\log$ term instead of just 2 and $\log t$.

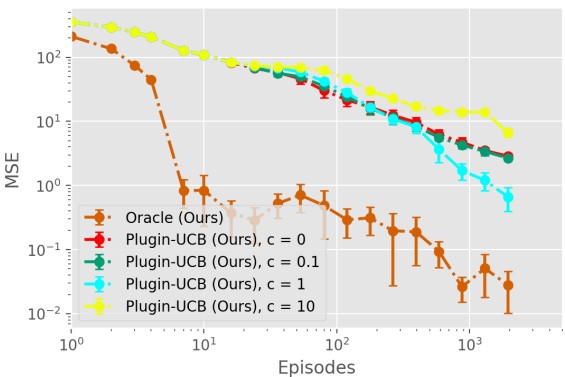

Figure 6: Ablation study of UCB constant

## J.3 ABLATION STUDY

In this experiment we show an ablation study of different values of the upper confidence bound constant associated with $\widehat{\sigma u}_t^k(s,a)$. Recall from (9) that

$$\widehat{\sigma u}_t^k(s_i^\ell, a) := \widehat{\sigma}_t^k(s_i^\ell, a) + 2c\sqrt{\frac{\log(SAn(n+1)/\delta)}{T_t^k(s_i^\ell, a)}}$$

where, $c$ is the upper confidence bound constant, and $n = KL$. From Theorem 2 we know that the theoretically correct constant is to use $2\eta + 4\eta^2$. However, since our upper bound is loose because of union bounds over states, actions, episodes and horizon, we ablate the value of $c$ to see its impact on ReVar . From Figure 6 we see that too large a value of $c = 10$ and we end up doing too much exploration rather than focusing on the state-action pair that reduces variance. However, even with too small values of $c \in \{0, 0.1\}$ we end up doing less exploration and have very bad plug-in estimates of the variance. Consequently this increases the MSE of ReVar . The value $c = 1$ seems to do relatively well against all the other choices.

# K   TABLE OF NOTATIONS

| Notations | Definition |
|---|---|
| $s_i^\ell$ | State $s$ in level $\ell$ indexed by $i$ |
| $\pi(a\|s_i^\ell)$ | Target policy probability for action $a$ in $s_i^\ell$ |
| $b(a\|s_i^\ell)$ | Behavior policy probability for action $a$ in $s_i^\ell$ |
| $\sigma^2(s_i^\ell, a)$ | Variance of action $a$ in $s_i^\ell$ |
| $\widehat{\sigma}_t^{(2),k}(s_i^\ell, a)$ | Empirical variance of action $a$ in $s_i^\ell$ at time $t$ in episode $k$ |
| $\widehat{\sigma u}_t^{(2),k}(s_i^\ell, a)$ | UCB on variance of action $a$ in $s_i^\ell$ at time $t$ in episode $k$ |
| $\mu(s_i^\ell, a)$ | Mean of action $a$ in $s_i^\ell$ |
| $\widehat{\mu}_t^k(s_i^\ell, a)$ | Empirical mean of action $a$ in $s_i^\ell$ at time $t$ in episode $k$ |
| $\mu^2(s_i^\ell, a)$ | Square of mean of action $a$ in $s_i^\ell$ |
| $\widehat{\mu}_t^{(2),k}(s_i^\ell, a)$ | Square of empirical mean of action $a$ in $s_i^\ell$ at time $t$ in episode $k$ |
| $T_n(s_i^\ell, a)$ | Total Samples of action $a$ in $s_i^\ell$ after $n$ timesteps |
| $T_n(s_i^\ell)$ | Total samples of actions in $s_i^\ell$ as $\sum_a T_n(s_i^\ell, a)$ after $n$ timesteps (State count) |
| $T_t^k(s_i^\ell, a)$ | Total samples of action $a$ taken till episode $k$ time $t$ in $s_i^\ell$ |
| $T_t^k(s_i^\ell, a, s_j^{\ell+1})$ | Total samples of action $a$ taken till episode $k$ time $t$ in $s_i^\ell$ to transition to $s_j^{\ell+1}$ |
| $P(s_j^{\ell+1}\|s_i^\ell, a)$ | Transition probability of taking action $a$ in state $s_i^\ell$ and transition to state $s_j^{\ell+1}$ |
| $\widehat{P}_t^k(s_j^{\ell+1}\|s_i^\ell, a)$ | Empirical transition probability of taking action $a$ in state $s_i^\ell$ and moving to state $s_j^{\ell+1}$ at time $t$ episode $k$ |
| $\widehat{P}_t^{(2),k}(s_j^{\ell+1}\|s_i^\ell, a)$ | Empirical square of transition probability of taking action $a$ in state $s_i^\ell$ and moving to state $s_j^{\ell+1}$ at time $t$ episode $k$ |
| $B(s_i^\ell) := \begin{cases} \sum_a \sqrt{\pi^2(a\|s_i^\ell)\sigma^2(s_i^\ell, a)}, \text{ if } \ell = L \\ \\ \sum_a \sqrt{\sum_{s_j^{\ell+1}} \pi^2(a\|s_i^\ell)\left(\sigma^2(s_i^\ell, a) + \gamma^2 P(s_j^{\ell+1}\|s_i^\ell, a)B^2(s_j^{\ell+1})\right)}, \text{ if } \ell \neq L \end{cases}$ | |
| $\widehat{B}(s_i^\ell) := \begin{cases} \sum_a \sqrt{\pi^2(a\|s_i^\ell)\widehat{\sigma}_t^{(2),k}(s_i^\ell, a)}, \text{ if } \ell = L \\ \\ \sum_a \sqrt{\sum_{s_j^{\ell+1}} \pi^2(a\|s_i^\ell)\left(\widehat{\sigma}_t^{(2),k}(s_i^\ell, a) + \gamma^2 \widehat{P}_t^k(s_j^{\ell+1}\|s_i^\ell, a)\widehat{B}_t^{(2),k}(s_j^{\ell+1})\right)}, \text{ if } \ell \neq L \end{cases}$ | |

Table 1: Table of Notations