# OpenReview forum: "ReVar: Strengthening Policy Evaluation via Reduced Variance Sampling"
_auai.org/UAI/2022/Conference — UAI 2022 Poster_

### Official Review · Reviewer_x3Uj · 2022-03-24

**Q2(1) Originality/Novelty:** 3
**Q2(2) Significance/Impact:** 3
**Q2(3) Correctness/Technical Quality:** 3
**Q2(6) Clarity Of Writing:** 4
**Q6 Overall Score:** 7
**Q8 Confidence In Your Score:** 3

**Q1 Summary And Contributions:**

This paper studied how we should optimally collect data for policy evaluation in MDPs with MSE as the evaluation metric. A new reduced variance sampling algorithm is proposed to lower the MSE. A complete theoretical study is conducted as well.

**Q2 Assessment Of The Paper:**

More detailed information regarding each of these aspects is given below:

**Q2(4) Quality Of Experiments (Optional):**

2: Fair: The experimental evaluation is weak: important baselines are missing, or the results do not adequately support the main claims.

**Q2(5) Reproducibility:**

2: Fair: Key resources (e.g., proofs, code, data) are unavailable but key details (e.g., proof sketches, experimental setup) are sufficiently well-described for an expert to confidently reproduce the main results.

**Q3 Main Strengths:**

This paper provided a fresh view of improving policy evaluation. As far as I know, most existing works focus on improving how to better compute the policy value given a pre-collected dataset. Certain conditions are added to those data as well as behavior policy. But not much work studied how to build a better behavior policy. From this point of view, this paper is novel. The theoretical analysis is solid.

**Q4 Main Weakness:**

I think the authors may want to emphasize the motivation a bit more. If policy evaluation is used in the offline RL, in most of the application, we do not have the control on the behavior policy. If policy evaluation is used as part of the RL algorithm such as policy iteration, I think MSE is not the ultimate goal. The goal here is to learn an optimal policy. Does it mean we have to maintain a separate policy to collect data at each iteration?

Although this is a solid study, the problem itself is a bit narrow since the MDP is not only tabular, but also tree-structured. The policy evaluation estimator is limited to one kind. Is the theory agnostic to different policy evaluation estimators? This proof is a bit long and tedious. It's better to highlight any technical contribution if any.

In this intro, you mentioned the on-policy method is far from optimal. This is fascinating for me. But I do not find a formal argument for that.


**Q5 Detailed Comments To The Authors:**

See above

**Q7 Justification For Your Score:**

This paper provided a new view to improve policy evaluation in terms of MSE. The theory is soild.

**Q9 Complying With Reviewing Instructions:**

1: Yes.

---

### Official Review · Reviewer_Xnsz · 2022-04-12

**Q2(1) Originality/Novelty:** 4
**Q2(2) Significance/Impact:** 3
**Q2(3) Correctness/Technical Quality:** 3
**Q2(6) Clarity Of Writing:** 4
**Q6 Overall Score:** 8
**Q8 Confidence In Your Score:** 3

**Q1 Summary And Contributions:**

This paper proposes a new method to reduce the estimation variance of policy evaluation. A theoretical guarantee is given and proofed for minimal-MSE of Tree MDP and possible extension to DAG MDP as well.

**Q2 Assessment Of The Paper:**

More detailed information regarding each of these aspects is given below:

**Q2(4) Quality Of Experiments (Optional):**

3: Good: The experimental evaluation is adequate, and the results convincingly support the main claims.

**Q2(5) Reproducibility:**

3: Good: Key resources (e.g., proofs, code, data) are available and key details (e.g., proofs, experimental setup) are sufficiently well-described for competent researchers to confidently reproduce the main results.

**Q3 Main Strengths:**

1.	The problem is clearly formalized and the paper is organized by starting from a simple setting multi-armed bandits to more complex one Tree MDP followed by a more general DAG MDP.
2.	Tree MDP is well defined and Figure 2 is helpful;
3.	The proposed idea is reasonable with sufficient details and proofs (even I did not check all the proofs in Appendix).
4.	Two toy examples are given to verify the idea empirically.


**Q4 Main Weakness:**

Even DAG MDP is more general than Tree MDP and multi-armed bandits, it is still a special kind of MDP and still with a gap to the commonly used general one.

**Q5 Detailed Comments To The Authors:**

Is that safe to directly use the results on the general MDP?

The $b$ here is more like the belief used in Bayesian RL. Is there any connection or relationship between them?

Are the results dependent on the distribution of reward function and dynamic transition? What if they are multi-mode distributions?  Do the results here need any assumptions on them?

**Q7 Justification For Your Score:**

In general, this is good work with sufficient contribution to the problem with strong theoretical analysis. I just has minor questions on the general usage of the results.

**Q9 Complying With Reviewing Instructions:**

1: Yes.

---

### Official Review · Reviewer_ESbP · 2022-04-13

**Q2(1) Originality/Novelty:** 3
**Q2(2) Significance/Impact:** 3
**Q2(3) Correctness/Technical Quality:** 3
**Q2(6) Clarity Of Writing:** 3
**Q6 Overall Score:** 5
**Q8 Confidence In Your Score:** 3

**Q1 Summary And Contributions:**

This paper proposed an approach for policy evaluation with reduced variance sampling.

**Q2 Assessment Of The Paper:**

More detailed information regarding each of these aspects is given below:

**Q2(4) Quality Of Experiments (Optional):**

3: Good: The experimental evaluation is adequate, and the results convincingly support the main claims.

**Q2(5) Reproducibility:**

3: Good: Key resources (e.g., proofs, code, data) are available and key details (e.g., proofs, experimental setup) are sufficiently well-described for competent researchers to confidently reproduce the main results.

**Q3 Main Strengths:**

The regret guarantees are derived for policy evaluation.

**Q4 Main Weakness:**

While the results are good, the comparisons with other approaches would have helped.

**Q5 Detailed Comments To The Authors:**

1. This paper considers an approach for policy evaluation. However, reduced variance has been used in policy gradient setups where ideas could be used or borrowed. Comparison of variance reduction approaches for policy gradient would help.
2. Comparison of regret bound to optimistic sampling or Dirichlet sampling type approaches would be good to see how much variance reduction helps.
3. The paper assumes state and action sets to be finite, and thus the approach is not scalable. While RL approaches have been extended to parametrized policies, extension of approaches in this work to large spaces will add to the work.

**Q7 Justification For Your Score:**

See above.

**Q9 Complying With Reviewing Instructions:**

1: Yes.

---

### Decision · Program_Chairs · 2022-05-15

**Decision:**

Accept (Poster)

**Comment:**

Meta Review: The authors have made a convincing rebuttal for one reviewer with a borderline opinion and two other reviewers firmly recommend acceptance.  The authors are encouraged to address reviewer concerns and incorporate their well-argued rebuttal response on revision.